JCB Journal of Cell Biology

# Neuronal activity regulates the nuclear proteome to promote activity-dependent transcription

Wendy A. Herbst[1,2], Weixian Deng[2] , James A. Wohlschlegel[2], Jennifer M. Achiro[2] , and Kelsey C. Martin[2]

**The formation and plasticity of neuronal circuits relies on dynamic activity-dependent gene expression. Although recent work has revealed the identity of important transcriptional regulators and of genes that are transcribed and translated in response to activity, relatively little is known about the cell biological mechanisms by which activity alters the nuclear proteome of neurons to link neuronal stimulation to transcription. Using nucleus-specific proteomic mapping in silenced and stimulated neurons, we uncovered an understudied mechanism of nuclear proteome regulation: activity-dependent proteasome-mediated degradation. We found that the tumor suppressor protein PDCD4 undergoes rapid stimulus-induced degradation in the nucleus of neurons. We demonstrate that degradation of PDCD4 is required for normal activity-dependent transcription and that PDCD4 target genes include those encoding proteins critical for synapse formation, remodeling, and transmission. Our findings highlight the importance of the nuclear proteasome in regulating the activity-dependent nuclear proteome and point to a specific role for PDCD4 as a regulator of activity-dependent transcription in neurons.**

## Introduction

Stimulus-induced gene expression allows neurons to adapt their structure and function in response to a dynamically changing external environment (Yap and Greenberg, 2018; Holt et al., 2019; Gallegos et al., 2018). Activity-dependent transcription is critical to neural circuit function, from synapse formation during development (West and Greenberg, 2011; Flavell et al., 2006; Wayman et al., 2006; Polleux et al., 2007; Lin et al., 2008) to synaptic plasticity in the mature brain (Bloodgood et al., 2013; Chen et al., 2017; Ramanan et al., 2005; Tyssowski et al., 2018; Yap and Greenberg, 2018). Neuronal activity can regulate gene expression in the nucleus through chromatin modification and transcriptional regulation and in the cytoplasm through regulation of RNA localization, stability, and translation (Martin and Ephrussi, 2009). The extreme morphological polarity and compartmentalization of neurons pose a cell biological challenge to the activity-dependent regulation of gene expression: how are stimuli received at one site coupled to changes in gene expression within specific subcellular compartments? To produce activity-dependent changes in transcription, signals must be relayed from the site where the signal is received, at the synapse, to the nucleus. To better understand how neuronal activity is coupled with changes in transcription, we developed an assay to systematically identify activity-dependent changes in the nuclear proteome of neurons and thereby elucidate novel mechanisms by which neuronal activity alters the concentration of

specific proteins in the nucleus. To our knowledge, this study provides the first activity-dependent characterization of the nuclear proteome of neurons.

Neuronal activity can change the concentration of nuclear proteins via a variety of mechanisms, from nucleocytoplasmic shuttling of signaling proteins to synthesis and degradation of nuclear proteins (Ch'ng et al., 2012; Dieterich et al., 2008; Ma et al., 2014; Lin et al., 2008; Bayraktar et al., 2020; Upadhya et al., 2004). Although the activity-dependent transcriptome and translatome of neurons have been extensively characterized (Brigidi et al., 2019; Chen et al., 2017; Dörrbaum et al., 2020; Fernandez-Albert et al., 2019; Hrvatin et al., 2018; Lacar et al., 2016; Tyssowski et al., 2018), less work has been focused on the upstream changes that occur in the nucleus, specifically identifying the population of proteins that undergo activity-dependent changes in nuclear abundance due to regulated transport or stability, which can in turn influence transcription. In this study, we used nucleus-specific proteomic mapping to identify preexisting proteins that undergo activity-dependent changes in concentration in the nucleus. By characterizing stimulus-induced changes in the nuclear proteome, these data provide insight into how activity-dependent transcription is regulated.

Through our screen of nuclear proteins with activity-dependent changes in abundance, we discovered that programmed cell

[1]Neuroscience Interdepartmental Program, University of California, Los Angeles, Los Angeles, CA;   [2]Department of Biological Chemistry, University of California, Los Angeles, Los Angeles, CA.

Correspondence to Kelsey C. Martin: kcmartin@mednet.ucla.edu;   Jennifer M. Achiro: jachiro@mednet.ucla.edu.



death 4 (PDCD4) undergoes a significant reduction in nuclear concentration following neuronal stimulation. PDCD4 has been studied primarily in the context of cancer, where it has been found to function as a tumor suppressor and translational inhibitor in the cytoplasm (Matsuhashi et al., 2019; Wang and Yang, 2018; Yang et al., 2003). These studies have revealed that the abundance of PDCD4 protein is regulated at multiple levels, including via translation (Asangani et al., 2008; Frankel et al., 2008; Ning et al., 2014), proteasome-mediated degradation (Dorrello et al., 2006), and nucleocytoplasmic trafficking (Böhm et al., 2003), with decreases in PDCD4 correlating with invasion, proliferation, and metastasis of many types of cancer (Allgayer, 2010; Chen et al., 2003; Wang and Yang, 2018; Wei et al., 2012).

Despite being expressed at significant levels in the brain, especially in the hippocampus and cortex (Lein et al., 2007; Li et al., 2021), few studies have addressed the role of PDCD4 in the nervous system. PDCD4 expression in neurons is altered by injury and stress (Jiang et al., 2017; Narasimhan et al., 2013; Li et al., 2021), and recent work has shown that, as in cancer cells, PDCD4 may act as a translational repressor in neurons (Li et al., 2021; Di Paolo et al., 2020). However, the impact of neuronal activity on PDCD4 concentration and the function of nuclear, as opposed to cytoplasmic, PDCD4 remains unknown. Here, we show that neuronal activity stimulates PKC- and proteasome-mediated degradation of PDCD4 in the nucleus of neurons and that PDCD4 degradation is necessary for activity-dependent transcription.

In this study, we describe an assay that represents the first, to our knowledge, to identify activity-dependent changes in the nuclear proteome of neurons, and it does so in a manner that is independent of translation. Our results not only elucidate a novel mechanism by which activity can regulate the nuclear proteome but also support a role for the tumor suppressor protein PDCD4 during activity-dependent transcription.

## Results

### Identification of the nuclear proteome from silenced and stimulated neurons using APEX2 proximity biotinylation and mass spectrometry (MS)

To identify proteins that undergo activity-dependent changes in nuclear localization or abundance, we analyzed the nuclear proteomes of silenced and stimulated cultured rat forebrain neurons. In developing this assay, we used cAMP-response element binding protein–regulated transcriptional coactivator 1 (CRTC1) as a positive control because we have previously shown that glutamatergic activity drives the synapse-to-nucleus import of CRTC1 and that neuronal silencing decreases CRTC1 nuclear abundance (Ch'ng et al., 2012, 2015). We initially used nuclear fractionation to capture the nuclear proteins but discovered that CRTC1 leaked out of the nucleus during the assay. This suggested that nuclear fractionation was not a suitable method and led us to instead use ascorbate peroxidase 2 (APEX2) proximity biotinylation (Hung et al., 2016), an in situ proximity ligation assay, to identify activity-dependent changes in the nuclear proteome. To specifically label the nuclear proteome, we

fused APEX2 to two SV40 NLSs (Kalderon et al., 1984; Fig. 1 A). APEX2 proximity ligation was advantageous for these experiments for the following reasons: (1) APEX2 biotinylated proteins can be captured directly by streptavidin pulldown, avoiding the need for subcellular fractionation; (2) biotinylation occurs rapidly (1-min labeling period); and (3) APEX2 can be expressed in a specific cell type of interest. We designed a neuron-specific nuclear-localized APEX2 construct (Fig. 1 A) and transduced cultured rat forebrain neurons with adeno-associated virus (AAV) expressing APEX2-NLS. Immunofluorescence of transduced neurons revealed that APEX2-NLS was expressed specifically in the nucleus (Fig. 1 B). We optimized the MOI of AAV to achieve high transduction efficiency without overexpression of the construct (important because higher doses of AAV led to APEX2-NLS expression in the cytoplasm). When all three components of the labeling reaction were supplied (APEX2-NLS, biotin-phenol, and $H_2O_2$), proteins were biotinylated specifically in neuronal nuclei (Fig. 1 B). No labeling was detected in the absence of APEX2-NLS, biotin-phenol, or $H_2O_2$.

To identify proteins that undergo activity-dependent changes in nuclear abundance, we silenced neurons for 1 h with the voltage-gated sodium channel antagonist tetrodotoxin (TTX) or stimulated neurons for 1 h with bicuculline (Bic), which inhibits $GABA_A$ receptors and drives glutamatergic transmission. We also inhibited protein synthesis using cycloheximide (CHX) in these experiments because many of the genes that are rapidly transcribed and translated in response to activity encode nuclear proteins (Yap and Greenberg, 2018; Heinz and Bloodgood, 2020; Alberini, 2009). We were concerned that the translation of activity-dependent genes would overshadow, and thereby hinder the detection of, the upstream changes that occur in the nuclear proteome resulting from alterations in nuclear protein localization or stability. We confirmed that the CHX treatment used throughout this study impairs protein translation, as detected by the strong reduction of new protein synthesis using L-azidohomoalanine (AHA) labeling (Fig. S1, A and B) as well as the complete inhibition of activity-dependent FOS induction (Fig. S1, C and D).

After the neurons were silenced or stimulated for 1 h and the 1-min labeling reaction was performed (Fig. 1 C), protein lysates were collected for analysis by Western blotting and MS. In neurons expressing APEX2-NLS, many proteins at a variety of molecular weights were biotinylated in both TTX and Bic conditions, whereas very few proteins were biotinylated in the no-APEX control, as detected by Western blot analysis (Fig. 1 D). The bands detected in the no-APEX control were at molecular weights of known endogenously biotinylated proteins (Hung et al., 2016). For all experiments, we also confirmed by immunocytochemistry (ICC) that the APEX2-NLS construct was localized to the nucleus during both experimental conditions (Fig. S1, E and F). For MS, biotinylated proteins were captured using streptavidin pulldown, and the nuclear proteomes were characterized using the tandem mass tag–MS/MS/MS–synchronous precursor selection (TMT-MS3-SPS) acquisition method (Ting et al., 2011) through liquid chromatography–MS. We detected 4,407 proteins, and, of those, 2,860 proteins were significantly enriched above the no-APEX negative control with $\log_2$ fold

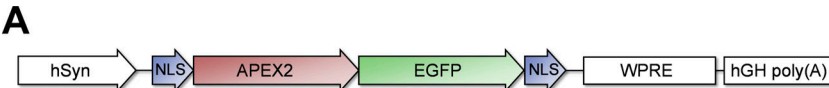

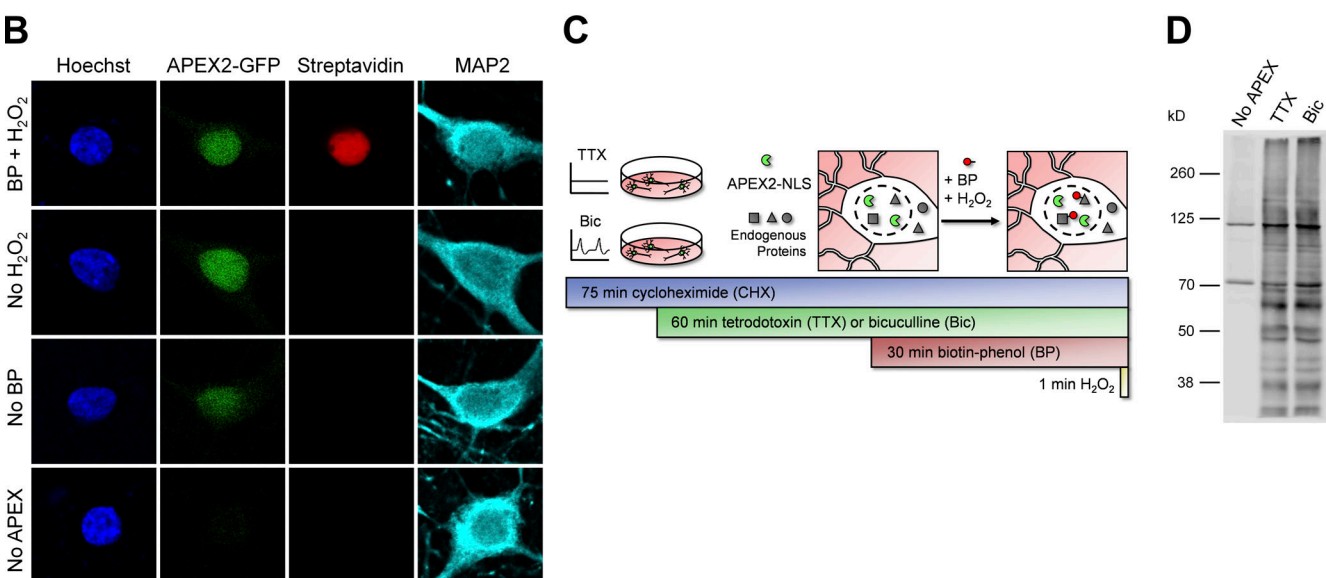

**E**

Top proteins enriched in nucleus of Bic-stimulated neurons:

| Symbol | Description | Log2FC Bic vs TTX | P-value |
|--------|-------------|-------------------|---------|
| CRTC1 | CREB Regulated Transcription Coactivator 1 | 1.21 | 0.0085 |
| TLK1 | Tousled Like Kinase 1 | 0.92 | 0 |
| COIL | Coilin | 0.79 | 0.0069 |
| PPWD1 | Peptidylprolyl Isomerase Domain And WD Repeat Containing 1 | 0.73 | 0.0016 |
| GON4L | Gon-4 Like | 0.73 | 0.0293 |

**F**

Top proteins enriched in nucleus of TTX-silenced neurons:

| Symbol | Description | Log2FC Bic vs TTX | P-value |
|--------|-------------|-------------------|---------|
| HDAC5 | Histone Deacetylase 5 | -1.04 | 0.0042 |
| HDAC4 | Histone Deacetylase 4 | -0.93 | 0.0024 |
| PDCD4 | Programmed Cell Death 4 | -0.89 | 0.0190 |
| TMEM147 | Transmembrane Protein 147 | -0.84 | 0.0430 |
| SMC5 | Structural Maintenance Of Chromosomes 5 | -0.82 | 0.0309 |

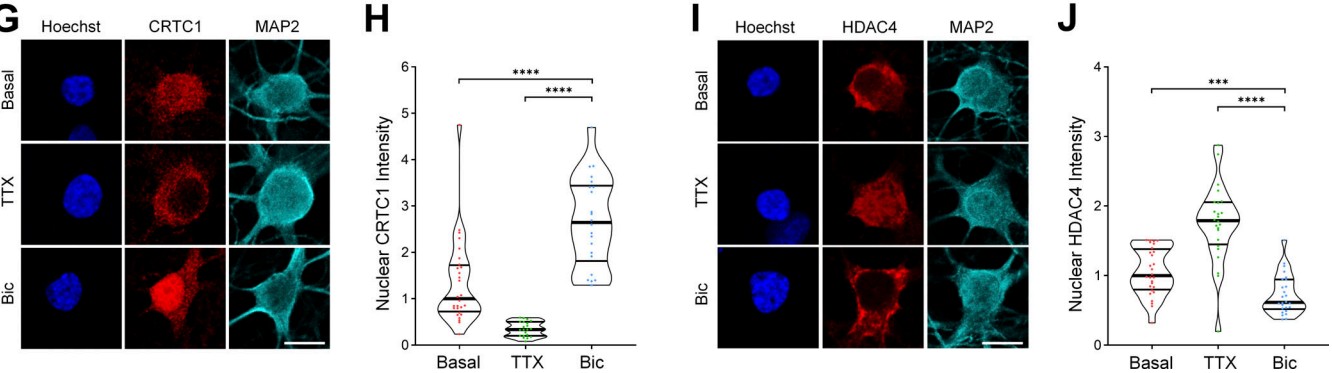

Figure 1. **Identification of the nuclear proteomes from silenced and stimulated neurons using APEX2 proximity biotinylation. (A)** Design of APEX2-NLS construct. hSyn, human synapsin promoter; NLS, SV40 NLS; WPRE, woodchuck hepatitis virus posttranscriptional regulatory element; hGH poly(A), human growth hormone polyadenylation sequence. **(B)** ICC of cultured rat forebrain neurons after APEX2 proximity biotinylation labeling. Nuclear proteins were biotinylated (streptavidin; red) by the combined presence of APEX2-NLS (GFP; green), biotin-phenol (BP), and $H_2O_2$. Scale bar, 10 µm. **(C)** Workflow for labeling nuclear proteins from silenced and stimulated neurons. APEX2 labeling diagram based on Hung et al. (2016). **(D)** Western blot of cultured neuron protein lysates from no-APEX, APEX+TTX, or APEX+Bic conditions, stained with streptavidin. **(E)** The top five proteins enriched in the nucleus of Bic-stimulated neurons, as ranked by Bic versus TTX $\log_2$ FC. **(F)** The top five proteins enriched in the nucleus of TTX-silenced neurons, as ranked by Bic versus TTX $\log_2$ FC. **(G)** CRTC1 ICC of basal, TTX-silenced, and Bic-stimulated neurons. Scale bar, 10 µm. **(H)** Violin plots of normalized nuclear CRTC1 ICC intensity. Basal, $n$ = 28;

TTX, $n$ = 21; and Bic, $n$ = 22 cells from 1 set of cultures. Basal median = 1.00; TTX median = 0.33; Bic median = 2.64. Basal versus Bic, P < 0.0001; TTX versus Bic, P < 0.0001. **(I)** HDAC4 ICC of basal, TTX-silenced, and Bic-stimulated neurons. Scale bar, 10 µm. **(J)** Violin plots of normalized nuclear HDAC4 ICC intensity. Basal, $n$ = 28; TTX, $n$ = 24; and Bic $n$ = 26 cells from 1 set of cultures. Basal median = 1.00; TTX median = 1.79; Bic median = 0.61. Basal versus Bic, P = 0.0006; TTX versus Bic, P < 0.0001. ***, P < 0.001; ****, P < 0.0001; Mann-Whitney $U$ test with Bonferroni correction.

change (FC) >3 and adjusted P value <0.05 (Table S1). To assess nuclear enrichment of the samples, we performed gene ontology (GO) cellular component analysis on the list of 2,860 proteins that were enriched above no-APEX (which could be matched to 2,615 unique gene identifiers). Approximately 80% of the top 100 most abundant proteins detected in the study contained the GO term "nucleus," and >60% of the entire list of 2,615 proteins contained the GO term nucleus (Fig. S1 G). We performed GO cellular component enrichment analysis and found that most of the top enriched terms were components of the nucleus (Fig. S1 H and Table S2).

When comparing the Bic and TTX conditions, 23 proteins were differentially expressed in the nucleus with $\log_2$ FC >0.5 or $\log_2$ FC less than –0.5 and P value <0.05 (Table S1). The highest-ranked protein by $\log_2$ FC enriched in the Bic versus TTX nuclear proteome was the synapse-to-nucleus signaling protein, CRTC1 (Ch'ng et al., 2015; Nonaka et al., 2014; Ch'ng et al., 2012; Sekeres et al., 2012; Fig. 1 E). The highest-ranked proteins enriched in the TTX versus Bic nuclear proteome were histone deacetylase 4 (HDAC4) and HDAC5, both of which have been reported to undergo nuclear export following neuronal stimulation (Chawla et al., 2003; Schlumm et al., 2013; Fig. 1 F). Using ICC, we confirmed that CRTC1 increased in the nucleus and HDAC4 decreased in the nucleus following Bic stimulation (Fig. 1, G–J). These findings demonstrate that the nucleus-specific APEX2 proximity ligation assay successfully captured activity-dependent changes in the nuclear proteome of neurons.

## Neuronal stimulation decreases PDCD4 protein concentration in the nucleus and cytoplasm of neurons

Among the proteins that underwent activity-dependent changes in nuclear concentration, the PDCD4 protein was significantly enriched in the TTX-treated nuclear proteome compared with the Bic-treated nuclear proteome. To validate this finding, we characterized the expression of PDCD4 protein in cultured neurons using ICC, and found that PDCD4 was present in both the nucleus and cytoplasm of neurons (Fig. 2 A and Fig. S2 A). Bic stimulation significantly decreased PDCD4 protein expression in the nucleus by ∼50% (Fig. 2, A and B), with a smaller decrease of ∼20% in the cytoplasm (Fig. 2 C). The decrease of PDCD4 occurred within 15 min of Bic stimulation, and PDCD4 protein levels continued to decrease further with longer incubations of Bic up to 3 h (Fig. 2 D). After washout of a 1-h Bic stimulation, PDCD4 protein expression gradually returned toward baseline levels, although PDCD4 protein concentration was still below baseline 24 h after washout (Fig. 2 E).

In complementary experiments, we transduced neurons with AAV expressing C-terminal HA-tagged PDCD4 (Fig. S2 B) and characterized PDCD4-HA expression by Western blot analysis (Fig. 2 F) and ICC (Fig. S2, C and D). By Western blot analysis, we found that total PDCD4-HA protein levels decreased by ∼50%

following Bic stimulation (Fig. 2 F). By ICC, both nuclear and cytoplasmic PDCD4-HA decreased by ∼40% (Fig. S2, C and D). To further validate the Bic-induced decrease in PDCD4, we also created an N-terminal V5-tagged PDCD4 plasmid (Fig. S2 B) and transfected the construct in neurons. Consistent with the results from endogenous PDCD4 and transduced C-terminally HA-tagged PDCD4, both nuclear and cytoplasmic V5-PDCD4 decreased by ∼40% with Bic stimulation (Fig. S2, E and F). These results show that exogenously expressed PDCD4 concentration in neurons is regulated by activity and that the decrease in PDCD4 represents a decrease in the protein rather than cleavage into multiple fragments.

We also found that depolarization of neurons with 40 mM KCl for 5 min significantly decreased PDCD4 protein concentration by ∼40% in the nucleus (Fig. 2 G) and ∼30% in the cytoplasm (Fig. 2 H), as detected immediately after the 5-min treatment. Altogether, these results indicate that increases in glutamatergic transmission and depolarization lead to a rapid and long-lasting reduction in PDCD4 abundance in the nucleus and, to a lesser extent, the cytoplasm of neurons.

## PDCD4 undergoes proteasome-mediated degradation following neuronal stimulation

In nonneuronal cells, PDCD4 has been reported to undergo miRNA-mediated translational repression (Asangani et al., 2008; Frankel et al., 2008; Ning et al., 2014), stimulus-induced nuclear export (Böhm et al., 2003), and stimulus-induced proteasome-mediated degradation (Dorrello et al., 2006). We sought to investigate the mechanisms of the neuronal activity-dependent decrease of PDCD4 (Fig. 3 and Fig. S3). Because the assay we used to detect activity-dependent changes in the nuclear proteome was conducted in the presence of the protein synthesis inhibitor CHX, we considered it unlikely that the Bic-induced decrease in PDCD4 was due to miRNA-mediated translational repression. To confirm this, we conducted PDCD4 ICC of TTX-silenced and Bic-stimulated neurons in the presence or absence of CHX and found that CHX did not block the Bic-induced decrease in PDCD4 (Fig. S3, A and B), thereby ruling out a role for activity-dependent miRNA-mediated translational regulation.

To investigate the mechanism underlying the decrease in nuclear PDCD4, we tested if the decrease in nuclear abundance was due to activity-dependent increases in nuclear export. Nuclear export of PDCD4 is mediated by the nuclear export protein, CRM1, and sensitive to the nuclear export inhibitor, leptomycin B (LMB; Böhm et al., 2003). Long incubation with LMB successfully caused nuclear accumulation of PDCD4 in unstimulated neurons (Fig. S3 C); however, LMB was unable to prevent the activity-dependent decrease of nuclear PDCD4 following stimulation (Fig. 3 A and Fig. S3 D). This result demonstrates that regulated nuclear export is not required for the activity-dependent decrease of PDCD4.

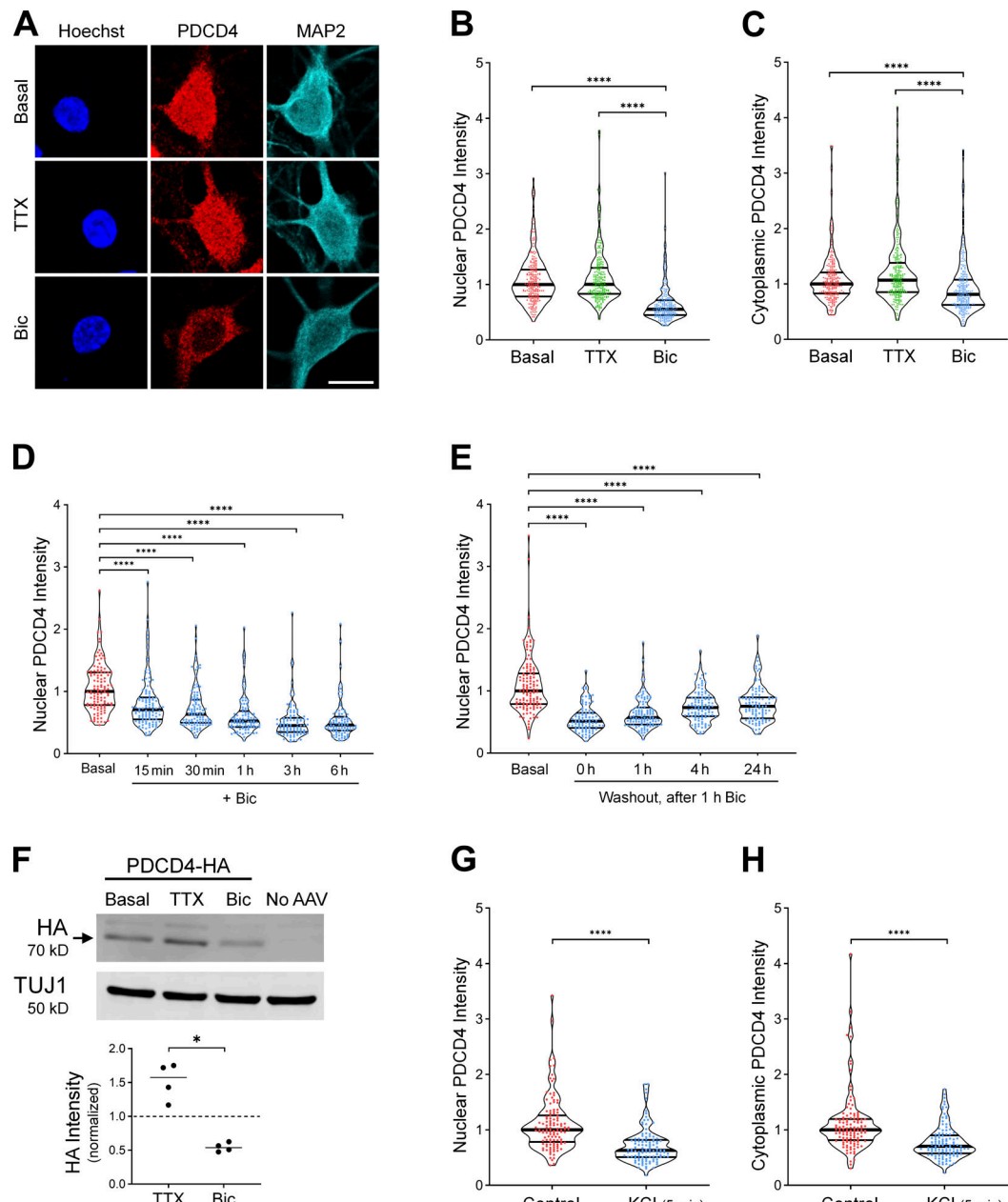

Figure 2. **Neuronal stimulation decreases PDCD4 protein concentration in the nucleus and cytoplasm of neurons. (A)** PDCD4 ICC of basal, TTX-silenced, and Bic-stimulated neurons. Scale bar, 10 μm. **(B)** Violin plots of normalized nuclear PDCD4 ICC intensity. Basal, $n$ = 226; TTX $n$ = 227; and Bic $n$ = 218 cells from 6 sets of cultures. Basal median = 1.00; TTX median = 1.00; Bic median = 0.55. Basal versus Bic, P < 0.0001; TTX versus Bic, P < 0.0001. **(C)** Violin plots of normalized cytoplasmic PDCD4 ICC intensity in the same cells as in B. Basal median = 1.00; TTX median = 1.07; Bic median = 0.81. Basal versus Bic, P < 0.0001; TTX versus Bic, P < 0.0001. **(D)** Violin plots of normalized nuclear PDCD4 ICC intensity after varying durations of Bic stimulation. Basal, $n$ = 115; Bic 15 min, $n$ = 111; Bic 30 min, $n$ = 102; Bic 1 h, $n$ = 97; Bic 3 h, $n$ = 92; and Bic 6 h, $n$ = 91 cells from three sets of cultures. Basal median = 1.00; Bic 15 min median = 0.71; Bic 30 min median = 0.63; Bic 1 h median = 0.52; Bic 3 h median = 0.45; Bic 6 h median = 0.46. Basal versus Bic 15 min, P < 0.0001; basal versus Bic 30 min, P < 0.0001; basal versus Bic 1 h, P < 0.0001; basal versus Bic 3 h, P < 0.0001; basal versus Bic 6 h, P < 0.0001. **(E)** Violin plots of normalized nuclear PDCD4 ICC intensity at various time points after removal of a 1-h Bic stimulation. Basal, $n$ = 124; washout 0 h, $n$ = 94; washout 1 h, $n$ = 111; washout 4 h, $n$ = 104; and washout 24 h, $n$ = 99 cells from 3 sets of cultures. Basal median = 1.00; washout 0 h median = 0.51; washout 1 h median = 0.57; washout 4 h median = 0.73; washout 24 h median = 0.75. Basal versus 0 h, P < 0.0001; basal versus 1 h, P < 0.0001; basal versus 4 h, P < 0.0001; basal versus 24 h, P < 0.0001. **(F)** Top: Western blot of protein lysates from basal, TTX-silenced, and Bic-stimulated neurons transduced with PDCD4-HA AAV. The PDCD4-HA band is indicated with an arrow. We observed a faint nonspecific band above the HA band. Bottom: Quantification of Western blot from four sets of cultures. HA intensity was normalized to TUJ1 intensity. Within each experiment, all samples were normalized to the basal sample. TTX/basal median = 1.57; Bic/basal median = 0.54. TTX versus Bic, P = 0.0286. **(G)** Violin plots of normalized nuclear PDCD4 ICC intensity. Control $n$ = 118 and KCl $n$ = 110 cells from three sets of cultures. Control median = 1.00; KCl median = 0.63. Control versus KCl, P < 0.0001. **(H)** Violin plots of normalized cytoplasmic PDCD4 ICC intensity in the same cells as in G. Control median = 1.00; KCl median = 0.70. Control versus KCl, P < 0.0001. *, P < 0.05; ****, P < 0.0001; Mann-Whitney $U$ test with Bonferroni correction.

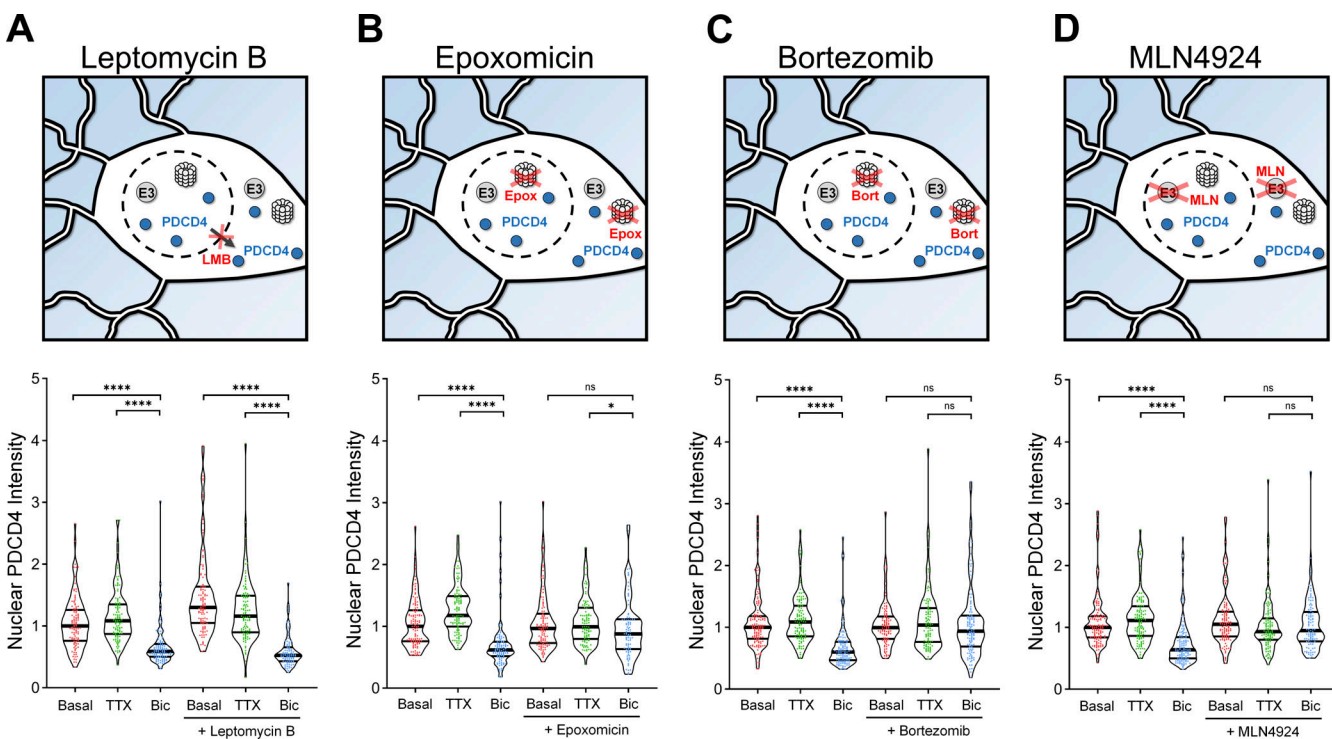

Figure 3. **PDCD4 undergoes proteasome-mediated degradation, not nuclear export, following neuronal stimulation. (A)** Top: Schematic of CRM1-mediated nuclear export inhibitor LMB experiments. Bottom: Violin plots of normalized nuclear PDCD4 ICC intensity. Basal, *n* = 109; TTX, *n* = 109; Bic, *n* = 109; LMB-basal, *n* = 106; LMB-TTX, *n* = 100; and LMB-Bic, *n* = 86 cells from 3 sets of cultures. Basal median = 1.00; TTX median = 1.08; Bic median = 0.59; LMB-basal median = 1.30; LMB-TTX median = 1.16; LMB-Bic median = 0.52. Basal versus Bic, P < 0.0001; TTX versus Bic, P < 0.0001; LMB-basal versus LMB-Bic, P < 0.0001; LMB-TTX versus LMB-Bic, P < 0.0001. **(B)** Top: Schematic of proteasome inhibitor Epox experiments. Bottom: Violin plots of normalized nuclear PDCD4 ICC intensity. Basal, *n* = 113; TTX, *n* = 106; Bic, *n* = 103; Epox-basal, *n* = 107; Epox-TTX, *n* = 94; and Epox-Bic, *n* = 86 cells from 3 sets of cultures. Basal median = 1.00; TTX median = 1.18; Bic median = 0.62; Epox-basal median = 0.97; Epox-TTX median = 0.99; Epox-Bic median = 0.88. Basal versus Bic, P < 0.0001; TTX versus Bic, P < 0.0001; Epox-basal versus Epox-Bic, P = 0.1662; Epox-TTX versus Epox-Bic, P = 0.0174. **(C)** Top: Schematic of proteasome inhibitor Bort experiments. Bottom: Violin plots of normalized nuclear PDCD4 ICC intensity. Basal, *n* = 131; TTX, *n* = 111; Bic, *n* = 120; Bort-basal, *n* = 100; Bort-TTX, *n* = 98; and Bort-Bic, *n* = 116 cells from three sets of cultures. Basal median = 1.00; TTX median = 1.09; Bic median = 0.60; Bort-basal median = 1.00; Bort-TTX median = 1.04; Bort-Bic median = 0.94. Basal versus Bic, P < 0.0001; TTX versus Bic, P < 0.0001; Bort-basal versus Bort-Bic, P = 0.6224; Bort-TTX versus Bort-Bic, P = 0.2648. **(D)** Top: Schematic of MLN experiments. MLN inhibits neddylation, preventing the activation of Cullin-RING E3 ubiquitin ligases. Bottom: Violin plots of normalized nuclear PDCD4 ICC intensity. Basal, *n* = 130; TTX, *n* = 120; Bic, *n* = 120; MLN-basal, *n* = 115; MLN-TTX, *n* = 108; and MLN-Bic, *n* = 97 cells from three sets of cultures. Basal median = 1.00; TTX median = 1.11; Bic median = 0.64; MLN-basal median = 1.05; MLN-TTX median = 0.93; MLN-Bic median = 0.95. Basal versus Bic, P < 0.0001; TTX versus Bic, P < 0.0001; MLN-basal versus MLN-Bic, P = 0.329; MLN-TTX versus MLN-Bic, P = 1. *, P < 0.05; ****, P < 0.0001; Mann-Whitney *U* test with Bonferroni correction.

We next hypothesized that regulated ubiquitin proteasome-mediated degradation may explain the Bic-induced decrease of PDCD4. To test this idea, we incubated TTX-silenced and Bic-stimulated neurons with the proteasome inhibitors epoxomicin (Epox) or bortezomib (Bort). The proteasome inhibitors impaired Bic-induced decreases of PDCD4 in both the nucleus and cytoplasm of neurons (Fig. 3, B and C; and Fig. S3, E and F), indicating that neuronal activity decreases PDCD4 concentrations via proteasome-mediated degradation. The finding that the nuclear export inhibitor LMB did not block the Bic-induced decrease of PDCD4 in the nucleus (Fig. 3 A) indicates that activity regulates proteasome-mediated degradation of nuclear PDCD4 directly in the nucleus, rather than by nuclear export of PDCD4 followed by degradation in the cytoplasm.

The E3 ubiquitin ligases β-transducin repeat-containing protein 1/2 (βTRCP1/2) have been shown to be required for the proteasome-mediated degradation of PDCD4 in the T98G glioblastoma cell line (Dorrello et al., 2006). βTRCP1/2 belong to

the family of Cullin-RING E3 ubiquitin ligases, which require neddylation in order to be activated (Merlet et al., 2009). To determine if this family of ligases is involved in the activity-dependent decrease of PDCD4 in neurons, we used the neddylation inhibitor MLN4924 (MLN) and found that it blocked the Bic-induced decrease of PDCD4 in the nucleus (Fig. 3 D) and cytoplasm (Fig. S3 G) of neurons. This result further supports the finding that PDCD4 undergoes proteasome-mediated degradation following Bic stimulation, likely through ubiquitination by βTRCP1/2.

### PDCD4 S71A mutation and PKC inhibition prevent the activity-dependent decrease of PDCD4

To understand how neuronal activity at synapses leads to proteasome-mediated degradation of PDCD4 in the nucleus, we focused on PDCD4's βTRCP-binding motif. Previous work in T98G glioblastoma cells has shown that phosphorylation of Ser67, Ser71, and/or Ser76 in the βTRCP-binding region of

PDCD4 is necessary for βTRCP binding and subsequent proteasome-mediated degradation of PDCD4 (Dorrello et al., 2006). To test if a mutation in PDCD4 at one of these sites would prevent the activity-dependent decrease of PDCD4 in neurons, we expressed WT and mutant (S71A) PDCD4-HA in cultured neurons using AAV (Fig. 4, A and B). Similar to endogenous PDCD4, we found that WT PDCD4-HA decreased following Bic stimulation, with an ~30% decrease in nuclear HA intensity and an ~15% decrease in cytoplasmic intensity, as detected by ICC (Fig. 4 C and Fig. S3 H). In contrast, the PDCD4-HA S71A mutant did not undergo an activity-dependent decrease in either the nucleus or cytoplasm, but showed a slight activity-dependent increase in nuclear intensity (Fig. 4 C). In complementary experiments, we found that Bic stimulation resulted in an ~40% decrease of WT PDCD4-HA signal by Western blot analysis, whereas PDCD4-HA S71A did not decrease after stimulation (Fig. 4 D). These results suggest that phosphorylation of S71 in PDCD4's βTRCP-binding motif is necessary for its activity-dependent degradation in neurons.

The βTRCP-binding region in PDCD4 contains canonical phosphorylation consensus sites for the kinases S6K1 and PKC (Fig. 4 B); Ser71 is a consensus site for PKC, and Ser67 and Ser76 are consensus sites for S6K1 (Dorrello et al., 2006; Matsuhashi et al., 2014, 2019; Nakashima et al., 2010). To test which kinase phosphorylates PDCD4 with neuronal activity, we incubated cultures in the S6K inhibitor LY2584702 or the PKC inhibitor Ro-31-8425. We found that the S6K inhibitor LY2584702 did not prevent the Bic-induced decrease of PDCD4 (Fig. 4 E and Fig. S3 I), despite its ability to potently inhibit the phosphorylation of a known S6K target, ribosomal protein S6 (Fig. S3 J). In contrast, the pan-PKC inhibitor Ro-31-8425 completely prevented the activity-dependent decrease (and slightly increased nuclear intensity) of PDCD4 (Fig. 4 F and Fig. S3 K). These results suggest that in response to glutamatergic activity, PKC phosphorylates PDCD4 at Ser71, which enables the ubiquitin ligase βTRCP1/2 to bind and promote the proteasome-mediated degradation of PDCD4.

## Stimulus-induced degradation of PDCD4 regulates the expression of neuronal activity-dependent genes

The finding that synaptic activity dynamically regulates PDCD4 protein concentration in the nucleus in response to synaptic activity points to a nuclear function for PDCD4 in coupling neuronal activity with changes in transcription. To investigate a role for PDCD4 in the regulation of activity-dependent transcription, we performed RNA sequencing (RNA-seq) of forebrain cultures transduced with either WT PDCD4 or degradation-resistant PDCD4 (S71A) following neuronal silencing with TTX or stimulation with Bic for 1 h (Fig. 5 and Table S3). Given that PDCD4 has a well-known role in regulating translation, we sought to distinguish between direct PDCD4-mediated transcriptional changes in the nucleus and the changes in expression that are downstream of PDCD4-mediated translational changes in the cytoplasm by performing the experiments in either the presence or absence of CHX. We detected robust Bic-induced increases in normalized read counts of transcripts for canonical immediate early genes such as *Npas4*, *Rgs2*, and *Egr4* in

all biological replicates (WT Bic versus TTX: *Npas4* $\log_2$ FC = 6.5; *Rgs2* $\log_2$ FC = 2.4; *Egr4* $\log_2$ FC = 3.1; Fig. 5 A and Table S3). We identified 912 activity-dependent genes, defined as genes with significant differential expression between Bic and TTX for PDCD4 WT samples (459 up-regulated, 453 down-regulated; adjusted P value <0.05 for WT no CHX; Fig. 5, B–D). Clustering of activity-dependent genes by FC across sample types revealed that most activity-dependent genes showed similar FCs between PDCD4 WT and PDCD4 S71A samples (Fig. 5 B), especially for genes with relatively high activity-dependent FCs (Fig. 5 C). However, Fig. 5 B also shows that PDCD4 S71A altered activity-dependent changes in gene expression for a subset of genes. Specifically, we found that PDCD4 S71A led to a decrease in activity-induced differential expression for a substantial proportion of genes: 43% of activity-dependent up-regulated genes (198 genes) and 57% of activity-dependent down-regulated genes (260 genes) were not significantly up-regulated or down-regulated, respectively, in the PDCD4 S71A samples (Fig. 5 D). These results suggest that regulated degradation of PDCD4 is important for the expression of activity-dependent genes in neurons.

This inhibition of activity-dependent gene expression could be due to both a potential role for PDCD4 in transcriptional processes and secondary effects from PDCD4's regulation of translation of specific transcripts (Matsuhashi et al., 2019; Wang and Yang, 2018). To isolate effects at the transcription level, we focused on CHX-insensitive activity-dependent genes, that is, genes that showed activity-dependent differential expression in both the presence and absence of CHX (in WT, 459 genes after excluding 3 genes that showed differential expression in different directions with or without CHX; Fig. 6). We ranked CHX-insensitive genes by their change in activity-dependent FC between PDCD4 WT and PDCD4 S71A samples and identified 91 putative PDCD4 target genes that showed large differences in activity-dependent gene expression between WT and degradation-resistant PDCD4 samples (see Materials and methods; Fig. 6 A). We validated the effect of PDCD4 on activity-dependent gene expression with quantitative RT-PCR (RT-qPCR) for two of the genes with the largest change between PDCD4 WT and PDCD4 S71A (*Scd1* and *Thrsp*; Fig. S4 A). To confirm a role for PDCD4 in regulating gene expression, we tested the effect of PDCD4 knockdown and overexpression on the transcript abundance of these putative PDCD4 target genes. We found that PDCD4 siRNA knockdown of endogenous PDCD4 significantly up-regulated the expression of *Scd1* and *Thrsp* (Fig. S4 B), demonstrating that endogenous PDCD4 represses the expression of these target genes. Overexpression of PDCD4 using PDCD4 AAV had a smaller effect on the target genes, significantly down-regulating *Thrsp* but only slightly and nonsignificantly down-regulating *Scd1* (Fig. S4 C). These findings support a role for PDCD4 in gene expression regulation and demonstrate that endogenous levels of PDCD4 are sufficient to repress the expression of target genes.

We performed motif analysis of promoter sequences of the putative PDCD4 target genes and found motifs similar to those in promoters of other CHX-insensitive activity-dependent genes (e.g., AP-1/TRE, ATF/CRE, Sp1/Klf motifs; Fig. S5), suggesting that there was not a specific transcription factor motif associated

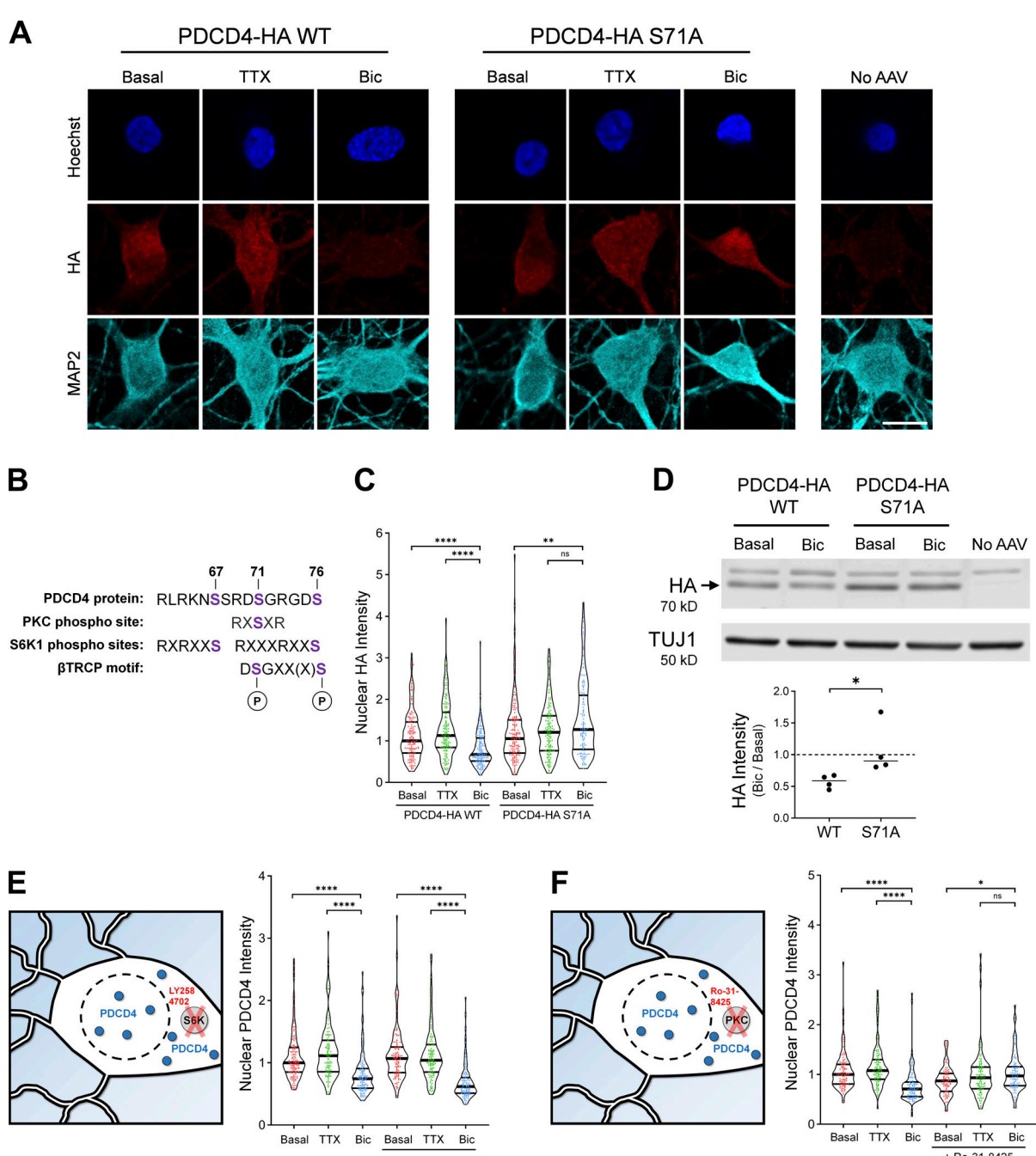

Figure 4. **PDCD4 S71A mutation and PKC inhibition prevent the activity-dependent decrease of PDCD4. (A)** HA ICC of basal, TTX-silenced, and Bic-stimulated neurons transduced with PDCD4-HA WT, PDCD4-HA S71A AAV, or no AAV (negative control). Scale bar, 10 μm. **(B)** PDCD4 protein sequence (aa 62–76). PKC and S6K1 phosphorylation sites are indicated in purple. Adapted from Matsuhashi et al. (2019). **(C)** Violin plots of normalized nuclear HA ICC intensity. WT-basal, n = 144; WT-TTX, n = 136; WT-Bic, n = 147; S71A-basal, n = 158; S71A-TTX, n = 140; and S71A-Bic, n = 122 cells from four sets of cultures. WT-basal median = 1.00; WT-TTX median = 1.13; WT-Bic median = 0.67; S71A-basal median = 1.05; S71A-TTX median = 1.21; S71A-Bic median = 1.27. WT-basal versus WT-Bic, P < 0.0001; WT-TTX versus WT-Bic, P < 0.0001; S71A-basal versus S71A-Bic, P = 0.0034; S71A-TTX versus S71A-Bic, P = 0.14. **(D)** Top: Western blot of protein lysates from basal and Bic-stimulated neurons transduced with PDCD4-HA WT or PDCD4-HA S71A. The PDCD4-HA band is indicated with an arrow. We observed a nonspecific band above the HA band. Bottom: Quantification of Western blot from four sets of cultures. HA intensity was normalized to TUJ1 intensity. Within each experiment, each Bic sample was normalized to its respective basal sample. WT Bic/Basal median = 0.59; S71A Bic/basal median = 0.90. WT versus S71A, P = 0.0286. **(E)** Left: Schematic of S6K inhibitor Ly2584702 (LY) experiments. Right: Violin plots of normalized nuclear PDCD4 ICC intensity. Basal, n = 138; TTX, n = 104; Bic, n = 122, LY-basal, n = 112; LY-TTX, n = 112; and LY-Bic, n = 107 cells from 3 sets of cultures. Basal median = 1.00; TTX median = 1.11; Bic median = 0.74; LY-basal median = 1.07; LY-TTX median = 1.04; LY-Bic median = 0.62. Basal versus Bic, P < 0.0001; TTX versus Bic, P < 0.0001; LY-basal versus LY-Bic, P < 0.0001; LY-TTX versus LY-Bic, P < 0.0001. **(F)** Left: Schematic of PKC inhibitor Ro-31-8425 (Ro) experiments. Right: Violin plots of normalized nuclear PDCD4 ICC intensity. Basal, n = 101; TTX, n = 101; Bic, n = 96; Ro-basal, n = 81; Ro-TTX, n = 95; and Ro-Bic, n = 87 cells from three sets of cultures. Basal median = 1.00; TTX median = 1.08; Bic median = 0.71; Ro-basal median = 0.87; Ro-TTX median = 0.93; Ro-Bic median = 0.97. Basal versus Bic, P < 0.0001; TTX versus Bic, P < 0.0001; Ro-basal versus Ro-Bic, P = 0.0114; Ro-TTX versus Ro-Bic, P = 0.3892. *, P < 0.05; **, P < 0.01; and ****, P < 0.0001; Mann-Whitney U test with Bonferroni correction.

Figure 5. **Stimulus-induced degradation of PDCD4 regulates the expression of neuronal activity–dependent genes. (A)** For each biological replicate, normalized read counts (from DESeq2) divided by transcript length are shown for the top 20 activity-dependent genes, ranked by adjusted P value for PDCD4 WT no-CHX samples. Each row represents a gene, and each column represents a biological replicate. The color of each box indicates transcript abundance (note: color is not scaled linearly in order to display full range of read counts; see Table S3 for full dataset). **(B)** Stimulation-induced log$_2$ FC for all 912 activity-dependent genes, clustered by FC across sample type. Each row represents a gene, and each column represents a sample type. The color legend represents Bic versus TTX log$_2$ FC, with red representing up-regulation, blue representing down-regulation, and white indicating log$_2$ FC of 0. **(C)** For each activity-dependent gene, Bic versus TTX log$_2$ FC is plotted against –log$_{10}$ of adjusted P value from PDCD4 WT samples (black) and PDCD4 S71A samples (red). Gene names for the top five activity-dependent genes by adjusted P value are labeled. For both PDCD4 WT and PDCD4 S71A samples, *Npas4* Bic versus TTX adjusted P value was 0 (–log$_{10}$ of 0 is not defined); therefore, for display, the –log$_{10}$ adjusted P value for *Npas4* was set to 250 for both samples. **(D)** Activity-dependent up-regulated genes (left bar) and activity-dependent down-regulated genes (right bar) were categorized by the activity-dependent differential expression in PDCD4 S71A samples. The colors in each bar show the percentage of activity-dependent genes showing activity-dependent up-regulation (red), no change (gray), or down-regulation (blue) in PDCD4 S71A samples.

with putative PDCD4 target genes. GO analysis of the putative PDCD4 target genes showed enrichment for neuronal signaling terms, such as "nervous system development" (GO:0007399; 28 genes; false discovery rate [FDR] = 9.28E-04) and "synapse" (GO: 0045202; 18 genes; FDR = 6.54E-03), whereas, for comparison, other CHX-insensitive activity-dependent genes showed enrichment for terms related to transcription, such as "regulation of gene expression" (GO:0010468; 169 genes; FDR = 3.11E-25) and "nuclear chromosome" (GO:0000228; 51 genes; FDR = 2.88E-09; Fig. 6 B and Table S4). Putative PDCD4 targets included genes

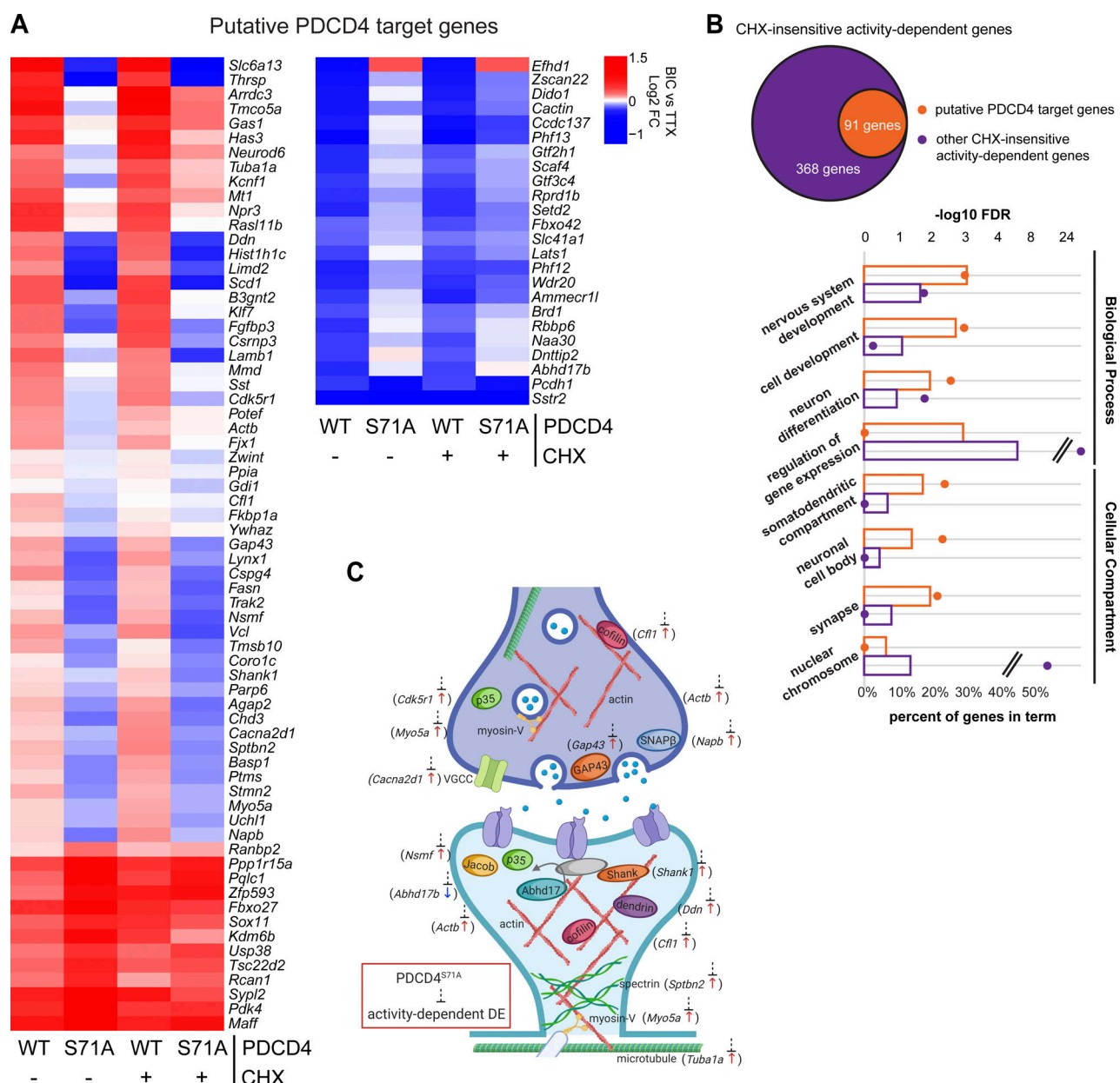

**Figure 6. Degradation-resistant PDCD4 suppresses activity-dependent changes in expression of synaptic genes. (A)** Bic versus TTX log$_2$ FC for all 91 putative PDCD4 transcriptional regulation target genes, clustered by FC across sample type. Each row represents a gene, and each column represents a sample type. The color legend represents Bic versus TTX log$_2$ FC, with red representing up-regulation, blue representing down-regulation, and white indicating a log$_2$ FC of 0. **(B)** GO analysis –log$_{10}$ FDR (circles) and percentage of genes (bars) in terms from biological process (top four terms) and cellular compartment (bottom four terms) analyses (Ashburner et al., 2000; Mi et al., 2019, The Gene Ontology Consortium, 2019). Data from putative PDCD4 target genes (91 genes) are shown in orange and, for comparison, data from other CHX-insensitive activity-dependent genes (368 genes) are shown in purple. Select GO terms are shown for simplicity (see Table S4 for top 15 GO terms by FDR for both groups of genes). **(C)** Diagram depicting a generic synapse and synaptic proteins. The labeled synaptic proteins are encoded by putative PDCD4 target genes (gene name indicated in parentheses alongside protein). The activity-dependent changes in expression of these genes are inhibited by degradation-resistant PDCD4. The presynaptic terminal is shown above with neurotransmitter-loaded synaptic vesicles, and the post-synaptic terminal is shown below with neurotransmitter receptors in the post-synaptic membrane (one receptor is shown anchored to an unlabeled gray PSD-95 protein). Arrow next to gene name illustrates the direction of activity-dependent differential expression, and dashed line with bar illustrates the suppression of this activity-dependent change in the PDCD4 S71A samples. DE, differential expression.

encoding proteins critical for synapse formation, remodeling, and transmission such as Shank1, p35, Abhd17b, Gap43, cofilin, spectrin-β2, myosin-Va, dendrin, Jacob, SNAP-β, voltage-dependent calcium channel-α2/δ1, α-tubulin, and β-actin (Fig. 6 C). Together, these results suggest that PDCD4 functions in the nucleus to regulate the expression of a subset of genes and that inhibiting the stimulation-induced degradation of nuclear PDCD4 results in suppression of the transcription of many activity-dependent genes important for neuronal synaptic function (Fig. 7).

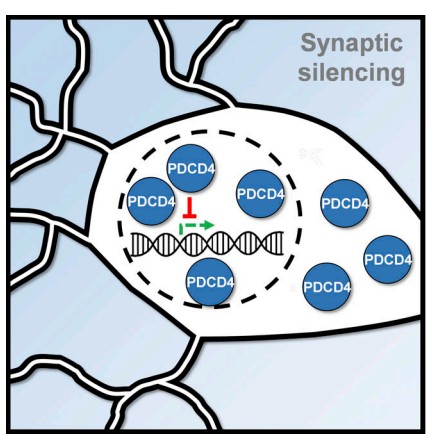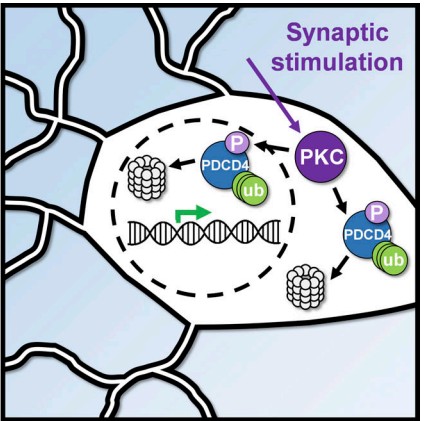

Figure 7. **Summary diagram of the activity-dependent proteasome-mediated degradation of PDCD4.** In silenced neurons (left), PDCD4 is highly expressed and suppresses the expression of specific genes. In stimulated neurons (right), PDCD4 is phosphorylated by PKC and undergoes proteasome-mediated degradation, thereby facilitating the expression of specific genes important for neuron synaptic function. ub, ubiquitin.

## Discussion

In this study, we sought to test how neuronal activity at the synapse changes the nuclear proteome to regulate activity-dependent transcription. While advances in transcriptomic technologies have enabled the identification of genes that undergo activity-dependent changes in expression (Brigidi et al., 2019; Chen et al., 2017; Fernandez-Albert et al., 2019; Tyssowski et al., 2018), the systematic identification of proteins that undergo changes in subcellular localization and/or stability has been more challenging. We implemented a new proximity ligation approach to characterize changes in the nuclear proteome triggered by neuronal stimulation. Our results provide the first, to our knowledge, unbiased characterization of the population of proteins that undergo changes in nuclear abundance following neuronal silencing and/or glutamatergic stimulation, and they do so in a manner that is independent of translation or transcription. We detected activity-dependent changes in the known nucleocytoplasmic shuttling proteins, CRTC1 and HDAC4/5 (Ch'ng et al., 2012; Chawla et al., 2003), demonstrating the validity of this neuron-specific, subcellular compartment–specific assay. In addition, our results highlight an understudied mechanism of transcriptional regulation in neurons: activity-dependent nuclear proteasome-mediated degradation.

We found that synaptic activity led to proteasome-mediated degradation of the tumor suppressor protein PDCD4 in the nucleus and, to a lesser extent, in the cytoplasm. Given that the nuclear export inhibitor LMB did not block the activity-dependent decrease of nuclear PDCD4, we demonstrated that nuclear PDCD4 is degraded without leaving the nucleus. Many examples of activity-dependent degradation of proteins within the cytoplasm have been reported in neurons (Hegde et al., 1993; Banerjee et al., 2009; Jarome et al., 2011), but fewer cases of activity-dependent degradation of proteins within the nucleus have been described (Upadhya et al., 2004; Bayraktar et al., 2020; Kravchick et al., 2016). Nonetheless, the nucleus contains machinery for proteasome-mediated degradation, and there are numerous examples of proteins that are degraded by the nuclear proteasome in nonneuronal cells, including transcriptional regulators and cell-cycle proteins (von Mikecz, 2006). Neuronal nuclei have also been shown to contain machinery for proteasome-mediated degradation (Mengual et al., 1996) and exhibit proteasomal activity, albeit with less activity

than is present in the cytoplasm (Upadhya et al., 2006; Tydlacka et al., 2008). Activity-dependent nuclear proteasome–mediated degradation of transcriptional regulators represents an important mechanism by which synaptic activity regulates gene expression.

In neurons, synaptic activity leads to an increase in the activity of many kinases, including S6K and PKC (Callender and Newton, 2017; Biever et al., 2015). We found that a phospho-incompetent serine-to-alanine mutation at Ser71 prevented the activity-dependent degradation of PDCD4 and that phosphorylation by PKC, but not S6K, was required for this activity-dependent degradation. This result is consistent with previous studies in other cell types, demonstrating that either S6K or PKC is required for PDCD4 phosphorylation, depending on the signaling pathway (Matsuhashi et al., 2019, 2014; Dorrello et al., 2006; Nakashima et al., 2010; Schmid et al., 2008). The stimulus-specific requirement of either S6K or PKC for PDCD4 degradation raises the interesting possibility that different types of neuronal stimulation could trigger PDCD4 degradation via distinct signaling pathways. Supporting this idea, two studies in neurons have suggested that PDCD4 degradation may be regulated by S6K in response to injury and stress (Li et al., 2021; Di Paolo et al., 2020), whereas our study demonstrated that PDCD4 degradation was mediated by PKC. PKC is typically activated at the cell surface (Gould and Newton, 2008); however, it can translocate from the cytoplasm to the nucleus after activation and can be activated directly in the nucleus (Lim et al., 2015; Martelli et al., 2006), where it phosphorylates nuclear targets, including histones and transcription factors (Lim et al., 2015; Martelli et al., 2006).

PDCD4 has been well characterized as a translational repressor in the cytoplasm of cancer cells (Wang et al., 2017; Wedeken et al., 2011; Yang et al., 2004). Low concentrations of PDCD4 have been reported to correlate with invasion, proliferation, and metastasis of many types of cancers (Allgayer, 2010; Chen et al., 2003; Wang and Yang, 2018; Wei et al., 2012). However, the role of PDCD4 in the nucleus is less well characterized, even though the protein is predominantly localized in the nucleus of many cells (Böhm et al., 2003). Despite being highly expressed in neurons, few studies have examined the neuronal function of PDCD4 (Di Paolo et al., 2020; Li et al., 2021; Narasimhan et al., 2013), and, as far as we are aware, no

previous study has identified a role for PDCD4 in activity-dependent transcription in neurons. In our study, we detected a larger activity-dependent decrease of PDCD4 in the nucleus than in the cytoplasm, and we found that blocking PDCD4 degradation suppressed activity-dependent gene expression. These findings suggest that the proteasome-mediated degradation of PDCD4 is important for regulating activity-dependent transcription following neuronal stimulation. Previous studies in nonneuronal cells have indicated that PDCD4 can inhibit AP-1–dependent transcription, although it is unclear whether this is a direct role in the nucleus (Bitomsky et al., 2004) or an indirect role regulating the translation of signaling proteins in the cytoplasm (Yang et al., 2006). PDCD4 has also been shown to bind to the transcription factors CSL (Jo et al., 2016) and TWIST1 (Shiota et al., 2009) and inhibit their transcriptional activity. Although activity-dependent degradation of cytoplasmic PDCD4 was weaker and less consistent than that of nuclear PDCD4, we cannot rule out the possibility that PDCD4 has a translation-independent role in the cytoplasm (such as binding to signaling proteins) that could indirectly regulate transcription. However, we propose that PDCD4 has a direct role in regulating activity-dependent transcription in the nucleus, in addition to its well-characterized role as a translational repressor in the cytoplasm.

We identified 91 genes that are putative activity-dependent targets of PDCD4, including genes encoding proteins that are important for synaptic function. The activity-dependent down-regulation of PDCD4 in neurons is reminiscent of the concept of "memory suppressor genes" (Abel and Kandel, 1998), genes that act as inhibitory constraints on activity-dependent neuronal plasticity. By analogy to PDCD4 function during cancer metastases, decreases in PDCD4 in neurons would function to enable experience-dependent neuronal growth and remodeling. Dysregulated PDCD4 concentrations have also been reported to underlie a variety of metabolic disorders, including polycystic ovary syndrome, obesity, diabetes, and atherosclerosis, highlighting the critical role PDCD4 plays in regulating gene expression in multiple cell types (Lu et al., 2020). Our study provides the first transcriptomic profile of PDCD4 that is independent of PDCD4's role in translation. These results provides insight into the transcriptional targets of PDCD4, which is of relevance not only to neuroscience but also to the study of PDCD4 in cancer.

Taken together, our findings illustrate the utility of proximity ligation assays in identifying activity-dependent changes in the proteome of subcellular neuronal compartments and point to the array of cell biological mechanisms by which activity can regulate the neuronal proteome. They also focus attention on the tumor suppressor protein PDCD4 as a critical regulator of activity-dependent gene expression in neurons, highlighting a role for PDCD4 in regulating the transcription of genes involved in synapse formation, remodeling, and transmission. This new role is in addition to PDCD4's well-characterized role as a translational inhibitor (Wang and Yang, 2018), and future investigation of the mechanisms by which PDCD4 regulates transcription of these genes will provide further insight into the understudied role of PDCD4 as a transcriptional regulator. Such studies also promise to deepen our understanding of the specific cell and molecular biological mechanisms by which experience alters gene expression in neurons to enable the formation and function of neural circuits.

## Materials and methods

### Experimental model details

#### Primary neuronal cultures

All experiments were performed using approaches approved by the University of California, Los Angeles, Animal Research Committee. Forebrains from postnatal day 0 Sprague-Dawley rats (Charles River) were dissected in cold HBSS (Thermo Fisher Scientific) supplemented with 10 mM Hepes buffer and 1 mM sodium pyruvate. Sex was not determined, and tissues from male and female pups were pooled. The tissue was chopped finely and digested in 1× trypsin solution (Thermo Fisher Scientific) in HBSS (supplemented with 120 µg/ml DNase and 1.2 mM $CaCl_2$) for 15 min at 37°C. The tissue was washed and triturated in DMEM (Thermo Fisher Scientific) + 10% FBS (Omega Scientific) before plating on polyDL-lysine (PDLL)-coated (0.1 mg/ml; Sigma) 10-cm dishes or 24-well plates containing acid-etched PDLL-coated coverslips (Carolina Biologicals). Neurons were plated at a density of one forebrain per 10-cm dish (for MS experiments) or one-half forebrain per entire 24-well plate (for ICC, RNA-seq, and Western blot experiments). Neurons were cultured in Neurobasal-A medium (Thermo Fisher Scientific) supplemented with 1× B-27 (Thermo Fisher Scientific), 0.5 mM GlutaMAX (Thermo Fisher Scientific), 25 µM monosodium glutamate (Sigma), and 25 µM β-mercaptoethanol (Sigma) and incubated at 37°C, 5% $CO_2$. When applicable, neurons were transfected with plasmids using Lipofectamine 2000 (Thermo Fisher Scientific) according to the manufacturer's instructions at day in vitro (DIV) 2, transduced with AAV at DIV 13, or treated with 1 µM control Accell siRNA (D-001910-10-20; Horizon Discovery) or 1 µM PDCD4 Accell siRNA (E-097927-00-0020; Horizon Discovery) at DIV 15. All experiments were performed at DIV 20.

### Detailed methods

#### Generation of plasmids and AAV

To create human synapsin promoter (hSyn) NLS-APEX2-EGFP-NLS, APEX2 was amplified from the pcDNA3 APEX2-NES plasmid (gift from Alice Ting, Stanford University, Stanford, CA; Addgene plasmid 49386) with three sequential sets of primers to add SV40 NLS to both the N-terminus and C-terminus of APEX2 (primer sets 1–3; Table S5). The design of using NLS on both sides of APEX2 was based on the design of Cas9-NLS (Swiech et al., 2015). NLS-APEX2-NLS was then inserted into the pAAV-hSyn-EGFP plasmid (gift from Bryan Roth, University of North Carolina, Chapel Hill, NC; Addgene plasmid 50465) between the BamHI and EcoRI sites, replacing the EGFP insert. The final hSyn NLS-APEX2-EGFP-NLS construct was created by amplifying hSyn NLS-APEX2-NLS (primer set 4; Table S5) and EGFP (primer set 5; Table S5) and joining the two products at NheI and SacI.

To create hSyn PDCD4-HA, rat PDCD4 was amplified from cultured neuron cDNA (primer set 6; Table S5) with a C-terminal

HA tag and then inserted into the pAAV-hSyn-EGFP plasmid between NcoI and EcoRI, replacing the EGFP insert. The S71A mutation was created using site-directed mutagenesis (services by GENEWIZ) to mutate serine 71 (TCT) to alanine (GCT). The hSyn V5-PDCD4-HA construct was created by adding a V5 tag to the N-terminus of PDCD4-HA using PCR-based mutagenesis (services by GENEWIZ).

AAV9 was generated for APEX2-NLS, PDCD4-HA WT, and PDCD4-HA S71A at the Penn Vector Core.

### Pharmacological treatments

Neurons were preincubated with CHX (60 µM; Sigma) for 15 min; LMB (10 nM; Sigma) for 30 min; LY2584702 (1 µM; Cayman Chemical) or Ro-31-8425 (5 µM; Sigma) for 1 h; or Epox (5 µM; Enzo Life Sciences), Bort (10 µM; APExBIO), or MLN (50 nM; APExBIO) for 2 h. Inhibitors remained in the media throughout the duration of each respective experiment, incubated at 37°C. For neurons treated with inhibitors dissolved in DMSO (Epox, Bort, MLN, LY2584702, and Ro-31-8425), the final DMSO concentration in the media was 0.1% or less. For neurons treated with LMB, the final methanol concentration in the media was 0.08%. All control groups received an equivalent concentration of vehicle (DMSO or methanol). To silence the neurons, 1 µM TTX (Tocris Bioscience) was applied to the neurons for 1 h. To stimulate the neurons, 40 µM (–)-Bic methiodide (Tocris Bioscience) was applied to the neurons for 1 h unless otherwise stated.

For KCl stimulations, neurons were preincubated with standard Tyrode's solution (140 mM NaCl, 10 mM Hepes, 5 mM KCl, 3 mM CaCl$_2$, 1 mM MgCl$_2$, and 20 mM glucose, pH 7.4) containing 1 µM TTX for 15 min at RT and then stimulated for 5 min with 40 mM KCl isotonic Tyrode's solution containing TTX. Control cells remained in the standard Tyrode's solution containing TTX throughout the experiment.

Control data from LMB (Fig. 3 A and Fig. S3 D) and CHX (Fig. S3, A and B) experiments were combined to generate the data shown in Fig. 2, B and C. Two of the Bort experiments were performed concurrently with two of the MLN experiments, so these experiments partially share control data in Fig. 3, C and D; and Fig. S3, F and G. One of the LY2584702 experiments was performed concurrently with one of the Bort experiments, so Fig. 4 E and Fig. S3 I partially share control data with Fig. 3 C and Fig. S3 F.

### APEX2 proximity biotinylation, streptavidin pulldown, and on-bead tryptic digestion

For APEX2 MS experiments, three biological replicates (sets of cultures) were prepared with three samples in each replicate (APEX + Bic, APEX + TTX, and no APEX). Neurons were TTX silenced or Bic stimulated for 1 h in the presence of CHX (as above). During the final 30 min of the treatment, neurons were incubated with 500 µM biotin-phenol (APExBIO) at 37°C. During the final 1 min, labeling was performed by adding H$_2$O$_2$ to a final concentration of 1 mM. To stop the labeling reaction, neurons were washed three times in large volumes of quencher solution (PBS with 10 mM sodium azide, 10 mM sodium ascorbate, and 5 mM Trolox [6-hydroxy-2,5,7,8-tetramethylchroman-2-carboxylic acid]).

Neurons were lysed with radioimmunoprecipitation assay (RIPA) buffer (50 mM Tris, 150 mM NaCl, 0.1% SDS, 0.5% sodium deoxycholate, and 1% Triton X-100, pH 7.5) containing protease inhibitor cocktail (Sigma), phosphatase inhibitor cocktail (Sigma), and quenchers. Lysates were treated with benzonase (200 U/mg protein; Sigma) for 5 min and then clarified by centrifugation at 15,000 $g$ for 10 min. Samples were concentrated with Amicon centrifugal filter tubes (3K NMWL; Millipore) to at least 1.5 mg/ml protein and quantified using a Pierce 660-nm protein assay kit (Thermo Fisher Scientific).

For each sample, 2 mg lysate was incubated with 220 µl Pierce streptavidin magnetic beads (Thermo Fisher Scientific) for 60 min at RT. Samples were washed twice with RIPA buffer; once with 1 M KCl; once with 0.1 M sodium carbonate; once with 2 M urea, and 10 mM Tris-HCl, pH 8.0; and twice with RIPA buffer according to Hung et al. (2016).

The streptavidin beads bound by biotinylated proteins were then washed three times with 8 M urea and 100 mM Tris-HCl, pH 8.5, and three times with pure water, and then the samples were resuspended in 100 µl 50 mM tetraethylammonium bromide. Samples were reduced and alkylated by sequentially incubating them with 5 mM tris(2-carboxyethyl)phosphine and 10 mM iodoacetamide for 30 min at RT in the dark on a shaker set to 1,000 rpm. The samples were incubated overnight with 0.4 µg Lys-C and 0.8 µg trypsin protease at 37°C on a shaker set to 1,000 rpm. Streptavidin beads were removed from peptide digests, and peptide digests were desalted using Pierce C18 tips (100-µl bead volume), dried, and then reconstituted in water.

### TMT labeling

The desalted peptide digests were labeled by the TMT10plex Isobaric Label Reagent Set (Thermo Fisher Scientific) according to the manufacturer's instructions. In brief, peptides were incubated with acetonitrile (ACN)-reconstituted TMT labeling reagent for 1 h and then quenched by adding hydroxylamine. Sample-label matches were no-APEX replicate 1 labeled with TMT126, APEX+Bic replicate 1 labeled with TMT127N, APEX+TTX replicate 1 labeled with TMT127C, APEX+TTX replicate 2 labeled with TMT128N, no-APEX replicate 2 labeled with TMT128C, no-APEX replicate 3 labeled with TMT129N, APEX+Bic replicate 2 labeled with TMT129C, APEX+Bic replicate 3 labeled with TMT130N, and APEX+TTX replicate 3 labeled with TMT130C. Labeled samples were then combined, dried, and reconstituted in 0.1% trifluoroacetic acid for high pH reversed-phase fractionation.

### High pH reversed-phase fractionation

High pH reversed-phase fractionation was performed using the Pierce High pH Reversed-Phase Peptide Fractionation Kit (Thermo Fisher Scientific) according to the manufacturer's instructions. In brief, peptides were bound to the resin in the spin column and then eluted by stepwise incubations with 300 µl of increasing ACN concentrations. The eight fractions were combined into four fractions (fractions 1 and 5, 2 and 6, 3 and 7, and 4 and 8). Fractions were then dried by vacuum centrifugation and reconstituted in 5% formic acid for MS analysis.

### Liquid chromatography–MS data acquisition

A 75-µm × 25-cm custom-made C18 column was connected to a nanoflow Dionex Ultimate 3000 ultra-high-pressure liquid chromatography system. A 140-min gradient of increasing ACN was delivered at a 200 nl/min flow rate as follows: 1–5.5% ACN phase from minutes 0 to 5, 5.5–27.5% ACN from minutes 5 to 128, 27.5–35% ACN from minutes 128 to 135, 35–80% ACN from minutes 135 to 136, 80% ACN hold from minutes 136 to 138, and then down to 1% ACN from minutes 138 to 140. An Orbitrap Fusion Lumos Tribrid mass spectrometer TMT-MS3-SPS method was used for data acquisition. Full MS scans were acquired at 120K resolution in the Orbitrap device with the automatic gain control target set to 2e5 and a maximum injection time set to 50 ms. MS2 scans were collected in ion trap with turbo scan rate after isolating precursors with an isolation window of 0.7 mass-to-charge ratio and collision-induced dissociation fragmentation using 35% collision energy. MS3 scans were acquired in the Orbitrap spectrometer at 50K resolution, and 10 synchronized selected precursor ions were pooled for each scan using 65% higher-energy C-trap dissociation energy for fragmentation. For data-dependent acquisition, a 3-s cycle time was used to acquire MS/MS spectra corresponding to peptide targets from the preceding full MS scan. Dynamic exclusion was set to 30 s.

### MS/MS database search

MS/MS database searching was performed using MaxQuant (1.6.10.43; Cox and Mann, 2008) against the rat reference proteome from the European Molecular Biology Laboratory (UP000002494-10116 RAT, *Rattus norvegicus*, 21,649 entries). The search included carbamidomethylation as a fixed modification and methionine oxidation and N-terminal acetylation as variable modifications. The digestion mode was set to trypsin and allowed a maximum of two missed cleavages. The precursor mass tolerances were set to 20 and 4.5 ppm for the first and second searches, respectively, whereas a 20-ppm mass tolerance was used for fragment ions. Datasets were filtered at 1% FDR at both the peptide spectrum match and protein levels. Quantification type was set to reporter ion MS3 with 10-plex TMT option.

### Statistical inference in MS data

MSStatsTMT (1.4.1; Huang et al., 2020) was used to analyze the MaxQuant TMT-MS3 data in the APEX2 proximity labeling experiment to statistically assess protein enrichment. TTX channels were used for MS run-level normalization. The "msstats" method was then used for protein summarization. P values for $t$ tests were corrected for multiple hypothesis testing using the Benjamini-Hochberg adjustment. We identified proteins that were enriched above the no-APEX negative control using a $\log_2$ FC >3 and adjusted P value <0.05 cutoff above the no-APEX condition. Using this protein list, we then identified proteins that were differentially expressed when comparing between Bic and TTX conditions using $\log_2$ FC >0.5 or $\log_2$ FC less than –0.5 and P value <0.05. It is important to note that we used a nonadjusted P value cutoff when identifying candidate proteins that were differentially expressed between the TTX-silenced and Bic-stimulated conditions, because only HDAC4 and six other proteins had a significant adjusted P value when using these cutoffs. Even CRTC1, a protein that has been shown to undergo activity-dependent changes in multiple studies (Ch'ng et al., 2012, 2015; Nonaka et al., 2014) and confirmed again in the present study, did not reach adjusted P value significance, suggesting that we did not have the statistical power to detect certain activity-dependent changes. Because we used nonadjusted P values to identify candidate proteins, it is especially important to experimentally validate any potential candidate protein.

### Protein extraction and Western blot analysis

Neurons were washed in Tyrode's solution (140 mM NaCl, 10 mM Hepes, 5 mM KCl, 3 mM CaCl$_2$, 1 mM MgCl$_2$, and 20 mM glucose, pH 7.4) and lysed with RIPA buffer (50 mM Tris, 150 mM NaCl, 0.1% SDS, 0.5% sodium deoxycholate, and 1% Triton X-100, pH 7.5) containing protease and phosphatase inhibitor cocktails (Sigma). Samples were clarified by centrifugation at 10,000 $g$ for 10 min. Protein concentration was determined using the Pierce bicinchoninic acid protein assay kit (Thermo Fisher Scientific).

Protein lysates were boiled in loading buffer (10% glycerol, 1% SDS, 60 mM Tris-HCl, pH 7.0, 0.1 M DTT, and 0.02% bromophenol blue) for 10 min at 95°C and run on an 8% polyacrylamide gel for 90 min at 120 V. Samples were wet transferred onto a 0.2-µm nitrocellulose membrane for 16 h at 40 mA. The membrane was blocked with Odyssey Blocking Buffer (LI-COR Biosciences) and incubated with the following primary antibodies: mouse HA (901513, 1:1,000; BioLegend), mouse TUJ1 (801201, 1:1,000; BioLegend), mouse S6 (2317, 1:1,000; Cell Signaling Technology), and rabbit phospho-S6 Ser235/236 (4858, 1:2,000; Cell Signaling Technology) for 3–4 h at RT or overnight at 4°C. The membrane was washed with TBS with Tween 20 and incubated with the following secondary antibodies: anti-rabbit IRDye 800CW (1:10,000), anti-mouse IRDye 800CW (1:10,000), anti-mouse IRDye 680CW (1:10,000), and IRDye 800CW streptavidin (1:1,000) for 1 h at RT. The membrane was imaged using the Odyssey infrared imaging system (LI-COR Biosciences). Western blots were quantified using the Image Studio (LI-COR Biosciences) rectangle tool. The relative intensity of each band was calculated by normalizing to a loading control (TUJ1). Within each experiment, all values were normalized to the control (basal) sample.

### ICC

Neurons were fixed with 4% PFA in PBS for 10 min, permeabilized in 0.1% Triton X-100 in PBS for 5 min, and blocked in 10% goat serum in PBS for 1 h. Neurons were incubated with the following primary antibodies: chicken microtubule-associated protein 2 (MAP2; 1100-MAP2, 1:1,000; PhosphoSolutions), rabbit PDCD4 (9535, 1:600; Cell Signaling Technology), mouse HA (901513, 1:1,000; BioLegend), mouse V5 (R960-25, 1:250; Thermo Fisher Scientific), rabbit CRTC1 (A300-769, 1:1,000; Bethyl Laboratories), rabbit HDAC4 (7628, 1:100; Cell Signaling Technology), and rabbit FOS (2250, 1:500; Cell Signaling Technology) for 3–4 h at RT or overnight at 4°C. Neurons were washed with

PBS and incubated at 1:1,000 with the following secondary antibodies: anti-chicken Alexa Fluor 647, anti-rabbit Alexa Fluor 555, anti-mouse Alexa Fluor 555, streptavidin Alexa Fluor 555, and Hoechst 33342 stain for 1 h at RT. Neurons were washed with PBS and mounted on slides with Aqua-Poly/Mount (Polysciences) for confocal imaging.

### Protein synthesis assay (AHA labeling)
Neurons were washed once in warm Tyrode's solution and then incubated in methionine-free Hibernate A (BrainBits) supplemented with 1× B-27 for 30 min at 37°C, 0% $CO_2$ (due to the buffering conditions of Hibernate A). Neurons were then incubated in 4 mM AHA (Thermo Fisher Scientific) or the nonlabeling control 4 mM methionine (Sigma) for 2 h in Hibernate A at 37°C, 0% $CO_2$. When applicable, 60 µm CHX was preincubated for 15 min before the start of the AHA labeling and remained in the media throughout the duration of the experiment. Neurons were washed twice in cold Tyrode's solution, fixed with 4% PFA in PBS for 10 min, permeabilized in 0.1% Triton X-100 in PBS for 5 min, and washed three times in 3% BSA in PBS. The Click-IT reaction was performed using the Click-iT Cell Reaction Buffer Kit (Thermo Fisher Scientific) and Alexa Fluor 488 alkyne (Thermo Fisher Scientific) according to the manufacturer's instructions, with a 30-min reaction at RT using 5 µM Alexa Fluor 488 alkyne. Neurons were washed three times in 3% BSA in PBS and then proceeded to the normal ICC protocol above, starting at the goat serum blocking step.

### Confocal imaging
Samples were imaged using a Zeiss LSM 700 confocal microscope with a 40×/1.3 NA oil objective and a 63×/1.4 NA oil objective at RT and 405-nm, 488-nm, 555-nm, and 639-nm lasers, using ZEN microscopy software. All images were acquired using the 40× objective, except for the AHA incorporation protein synthesis experiments, which were imaged using the 63× objective. Identical image acquisition settings were used for all images within an experiment. For each image acquisition, the experimenter viewed the MAP2 and Hoechst channels to select a field of view and was blind to the experimental channel (e.g., HA, PDCD4). For each coverslip, images were taken at multiple regions throughout the coverslip, and two or three coverslips were imaged per condition. Images were collected from at least three experimental replicates (sets of cultures), unless otherwise stated.

### Image analysis
ICC images were processed using ImageJ (Schindelin et al., 2012). An ImageJ macro was used to create regions of interest (ROIs) for neuronal nuclei. In brief, the Hoechst signal was used to outline the nucleus, and the MAP2 signal was used to select neurons and exclude nonneuronal cells. To create ROIs for neuronal cytoplasm, the cell body of each neuron was manually outlined using the MAP2 signal, and then the nuclear ROI was subtracted from the total cell body ROI. The ROIs were used to calculate the mean intensity in the channel of interest (e.g., HA, PDCD4) for the nucleus and cytoplasm of each neuron. Within each ICC experimental replicate, the measured values from all ROIs were normalized to the median value of the control condition (basal). For experiments using transfected cells (V5 experiments), the measured intensity of each ROI was normalized to the cotransfection marker (nuclear GFP intensity) in order to normalize for differences in transfection efficiency between cells.

### RNA extraction, library preparation, RNA-seq, and data analysis
Samples were prepared from three biological replicates (sets of cultures), with eight samples in each replicate (WT Bic, WT TTX, S71A Bic, S71A TTX, CHX WT Bic, CHX WT TTX, CHX S71A Bic, CHX S71A TTX). RNA was extracted from neuronal cultures using the RNeasy Micro Kit (Qiagen) according to the manufacturer's instructions. Libraries for RNA-seq were prepared with the NuGEN Universal Plus mRNA-Seq Kit (NuGEN) to generate strand-specific RNA-seq libraries. Samples were multiplexed, and sequencing was performed on an Illumina HiSeq 3000 system to a depth of 25 million reads per sample with single-end 65-bp reads. Demultiplexing was performed using Illumina Bcl2fastq v2.19.1.403 software. The RNA-seq data discussed in this publication have been deposited in the National Center for Biotechnology Information Gene Expression Omnibus (Edgar et al., 2002) and are accessible through Gene Expression Omnibus accession no. GSE163127. Reads were aligned to *Rattus norvegicus* reference genome version Rnor_6.0 (rn6), and reads per gene were quantified by STAR 2.27a (Dobin et al., 2013) using an Rnor_6.0 gtf file. We used DESeq2 (Love et al., 2014) to obtain normalized read counts and perform differential expression analysis, including batch correction for replicate number (Table S3). Putative PDCD4 target genes were identified by first focusing on genes that showed activity-dependent differential expression in both the presence and absence of CHX (CHX-insensitive activity-dependent genes; 459 genes after excluding 3 genes that showed differential expression in different directions with or without CHX). We then calculated the PDCD4 activity-dependent change by taking the difference between the activity-dependent FC of PDCD4 WT and PDCD4 S71A samples and normalizing:

$$PDCD4 \text{ change index } = \text{ abs}(S - W)/\text{abs}(W),$$

where $S$ is the PDCD4 S71A no-CHX Bic versus TTX $\log_2$ FC and $W$ is the PDCD4 WT no-CHX Bic versus TTX $\log_2$ FC. We defined putative PDCD4 target genes as those with a PDCD4 change index >0.75. Motif analysis was performed using the findMotifsGenome command in HOMER (Heinz et al., 2010), using sequences from the transcription start site and upstream 500 bp as the promoter sequences for each gene. GO analysis was performed using the GO resource (Ashburner et al., 2000, The Gene Ontology Consortium, 2019) and PANTHER enrichment tools (Mi et al., 2019). A cartoon of putative PDCD4 targets was generated using BioRender.com.

### RT-qPCR
As above, RNA was extracted from neuronal cultures using the RNeasy Micro Kit (Qiagen). cDNA was synthesized from RNA using the SuperScript III First-Strand Synthesis System (Thermo Fisher Scientific) with random hexamers. A no–reverse transcriptase sample was also prepared as a negative

control. RT-qPCR was performed on the CFX Connect Real-Time System (Bio-Rad Laboratories) using PowerUp SYBR Green Master Mix (Applied Biosystems). Primer pairs were designed for two housekeeping genes (*Hprt*, *Gapdh*), two candidate genes (*Scd1*, *Thrsp*), and *Pdcd4* using Primer3Plus (Untergasser et al., 2012) and National Center for Biotechnology Information Primer-BLAST (Ye et al., 2012; Table S5). RT-qPCR was performed on six or seven sets of cultures, with technical triplicate reactions for each sample. For each gene, relative quantity was calculated using the formula: $E^{\Delta Ct}$, where E was calculated from the primer efficiencies (E ≈ 2). Relative gene expression was calculated by normalizing the relative quantity of the gene of interest to the relative quantity of the housekeeping genes *Hprt* and *Gapdh*: $(E_{gene})^{\Delta Ct\ gene}/\text{GeoMean}[(E_{HPRT})^{\Delta Ct\ HPRT}, (E_{GAPDH})^{\Delta Ct\ GAPDH}]$ (for Fig. S4 A) or *Gapdh* only (for Fig. S4, B and C; $E_{gene})^{\Delta Ct\ gene}/(E_{GAPDH})^{\Delta Ct\ GAPDH}$.

### Quantification and statistical analysis

For ICC experiments, the quantification of signal intensity is displayed in violin plots created using GraphPad Prism. The medians are indicated with thick lines, and the quartiles are indicated with thin lines. *n* refers to the number of neurons in each condition, and all individual data points were plotted on the graphs. Our sample sizes were not predetermined. A nonparametric statistical test (Mann-Whitney *U* test) was used to calculate statistical significance because our data were not normally distributed, as indicated by the violin plots. A Bonferroni correction was used to adjust for multiple hypothesis testing. The sample sizes, statistical tests, medians, and P values are indicated in the figure legends.

For RT-qPCR and Western blot experiments, all data points were displayed using GraphPad Prism, with solid lines indicating the median values. *n* refers to the biological replicates (sets of cultures), and all data points were plotted on the graphs. The Mann-Whitney *U* test (Prism) was used to calculate statistical significance. The sample size, statistical tests, medians, and P values are indicated in the figure legends.

### Online supplemental material

Fig. S1 accompanies Fig. 1 and provides verification of the protein synthesis inhibitor CHX, examines the nuclear localization of APEX2-NLS, and provides GO analysis results. Fig. S2 accompanies Fig. 2 and provides ICC data of PDCD4 using antibodies against the endogenous protein and epitope-tagged PDCD4. Fig. S3 accompanies Figs. 3 and 4 and provides verification of the nuclear export inhibitor LMB and S6K inhibitor Ly2584702, as well as quantification of cytoplasmic PDCD4. Fig. S4 accompanies Fig. 6 and provides RT-qPCR validation of candidate genes from RNA-seq. Fig. S5 accompanies Fig. 6 and provides motif analysis of promoters of putative PDCD4 target genes. Table S1 contains the MS data from the nuclear proteomes of silenced and stimulated neurons. Table S2 lists the GO analysis of the APEX2-NLS MS. Table S3 contains the RNA-seq data from silenced and stimulated neurons, with PDCD4 WT or S71A. Table S4 lists the GO analysis for putative PDCD4 target genes. Table S5 lists the primer sequences used for cloning and RT-qPCR.

### Data availability

Further information and requests for resources and reagents should be directed to and will be fulfilled by the lead contact author, K.C. Martin (kcmartin@mednet.ucla.edu). Plasmids generated in this study are available upon request from the lead contact author. This published article includes the MS data generated during this study. The full RNA-seq data are available online (Gene Expression Omnibus accession no. GSE163127).

### Acknowledgments

We thank Sylvia Neumann, Marika Watanabe, and Emilie Marcus for their comments on the manuscript and members of the Martin laboratory for helpful discussion. RNA-seq was performed at the Technology Center for Genomics & Bioinformatics at the University of California, Los Angeles.

This work was supported by National Institutes of Health grant R01MH077022 to K.C. Martin and National Institutes of Health National Research Service Award F31MH113310 to W.A. Herbst.

The authors declare no competing financial interests.

Author contributions: Conceptualization: W.A. Herbst and K.C. Martin. Methodology: W.A. Herbst, W. Deng, J.A. Wohlschlegel, and K.C. Martin. Investigation: W.A. Herbst (APEX2 and PDCD4 neuron experiments), W. Deng (MS), and J.M. Achiro (RNA-seq data analysis). Writing – original draft: W.A. Herbst, J.M. Achiro, and K.C. Martin. Writing – review and editing: W.A. Herbst, J.M. Achiro, and K.C. Martin. Supervision: J.A. Wohlschlegel, J.M. Achiro, and K.C. Martin.

Submitted: 24 March 2021

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

# Supplemental material

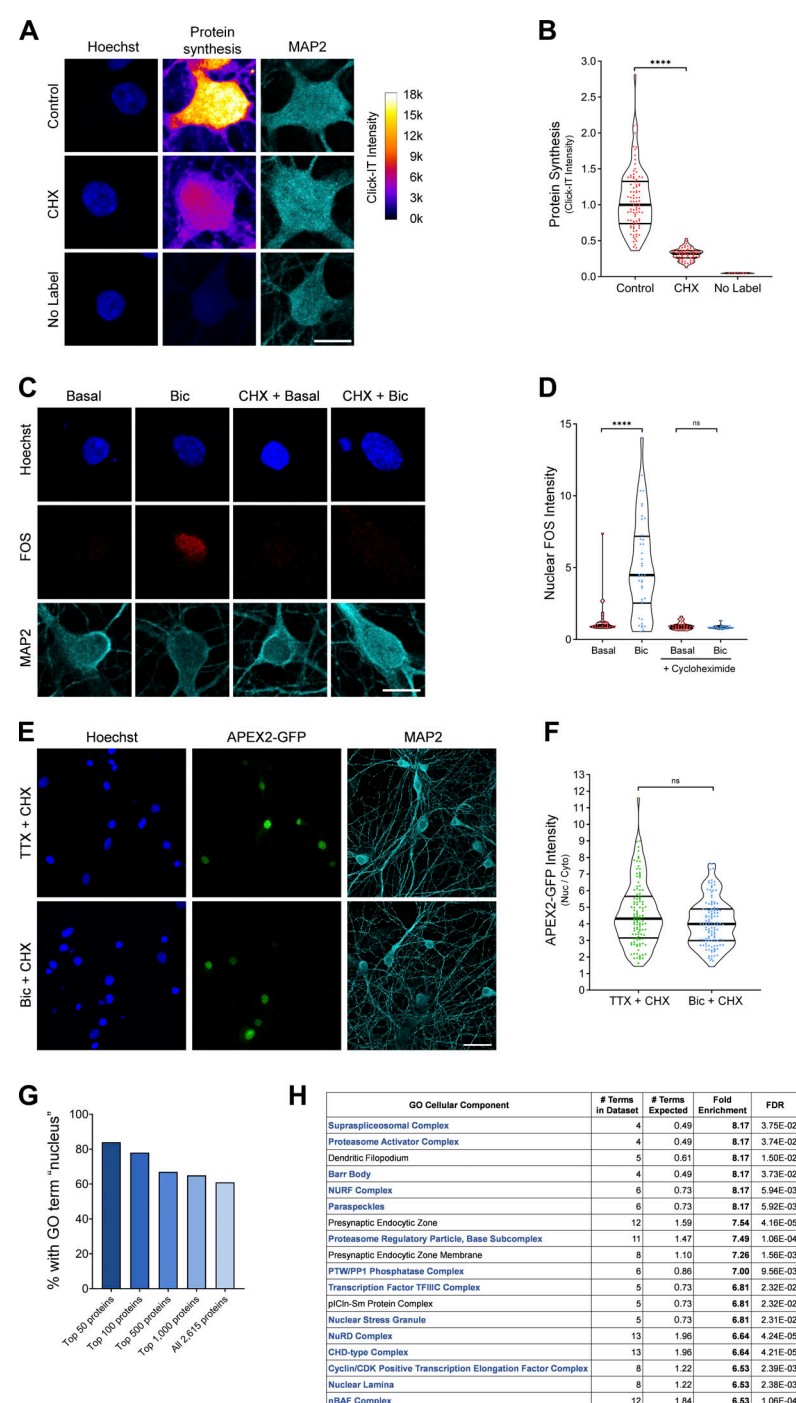

**H**

| GO Cellular Component | # Terms in Dataset | # Terms Expected | Fold Enrichment | FDR |
|---|---|---|---|---|
| Supraspliceosomal Complex | 4 | 0.49 | 8.17 | 3.75E-02 |
| Proteasome Activator Complex | 4 | 0.49 | 8.17 | 3.74E-02 |
| Dendritic Filopodium | 5 | 0.61 | 8.17 | 1.50E-02 |
| Barr Body | 4 | 0.49 | 8.17 | 3.73E-02 |
| NURF Complex | 6 | 0.73 | 8.17 | 5.94E-03 |
| Paraspeckles | 6 | 0.73 | 8.17 | 5.92E-03 |
| Presynaptic Endocytic Zone | 12 | 1.59 | 7.54 | 4.16E-05 |
| Proteasome Regulatory Particle, Base Subcomplex | 11 | 1.47 | 7.49 | 1.06E-04 |
| Presynaptic Endocytic Zone Membrane | 8 | 1.10 | 7.26 | 1.56E-03 |
| PTW/PP1 Phosphatase Complex | 6 | 0.86 | 7.00 | 9.56E-03 |
| Transcription Factor TFIIIC Complex | 5 | 0.73 | 6.81 | 2.32E-02 |
| pICln-Sm Protein Complex | 5 | 0.73 | 6.81 | 2.32E-02 |
| Nuclear Stress Granule | 5 | 0.73 | 6.81 | 2.31E-02 |
| NuRD Complex | 13 | 1.96 | 6.64 | 4.24E-05 |
| CHD-type Complex | 13 | 1.96 | 6.64 | 4.21E-05 |
| Cyclin/CDK Positive Transcription Elongation Factor Complex | 8 | 1.22 | 6.53 | 2.39E-03 |
| Nuclear Lamina | 8 | 1.22 | 6.53 | 2.38E-03 |
| nBAF Complex | 12 | 1.84 | 6.53 | 1.06E-04 |

Figure S1.  **Identification of the nuclear proteomes from silenced and stimulated neurons using APEX2 proximity biotinylation.** Related to Fig. 1. **(A)** Metabolic labeling and ICC of control, CHX-treated, and nonlabeled cells. Protein synthesis was measured by AHA incorporation and Click-IT Alexa Fluor 488 alkyne reaction. The Alexa Fluor 488 signal intensity is displayed using a lookup table. Scale bar, 10 µm. **(B)** Violin plots of normalized somatic Click-IT intensity. Control, $n$ = 90 cells; CHX, $n$ = 63 cells; and no label, $n$ = 7 cells from two sets of cultures. Control median = 1.00; CHX median = 0.32; no-label median = 0.05. Control versus CHX, P < 0.0001. **(C)** FOS ICC of basal and Bic-stimulated neurons in the presence or absence of CHX. Scale bar, 10 µm. **(D)** Violin plots of normalized nuclear FOS ICC intensity. Basal, $n$ = 28; Bic, $n$ = 40 cells; CHX-basal, $n$ = 32; and CHX-Bic, $n$ = 26 cells from one set of cultures. Basal median = 1.00; Bic median = 4.484; CHX-basal median = 0.8941; CHX-Bic median = 0.8307. Basal versus Bic, P < 0.0001; CHX-basal versus CHX-Bic, P = 0.3064. **(E)** Localization of APEX2-NLS GFP in TTX-silenced and Bic-stimulated neurons in the presence of CHX. Scale bar, 30 µm. **(F)** Violin plots of APEX2-NLS GFP nucleocytoplasmic localization (nuclear [Nuc]/cytoplasmic [Cyto] ratio) in TTX-silenced and Bic-stimulated neurons in the presence of CHX. TTX, $n$ = 116 cells; and Bic, $n$ = 120 cells from nine sets of cultures. TTX median = 4.31; Bic median = 4.00. TTX versus Bic, P = 0.0562. **(G)** Bar graph displaying the percentage of proteins containing the GO term nucleus (GO:0005634) from the list of proteins detected by APEX2-NLS MS. The list of proteins detected by MS was ranked according to MS/MS count abundance, and separate bars are displayed for the top 50 proteins, top 100 proteins, and so forth. **(H)** Cellular component GO analysis of the APEX2-NLS MS, with the top 18 terms listed in order of fold enrichment. Nuclear cellular components are listed in blue. ****, P < 0.0001; Mann-Whitney $U$ test.

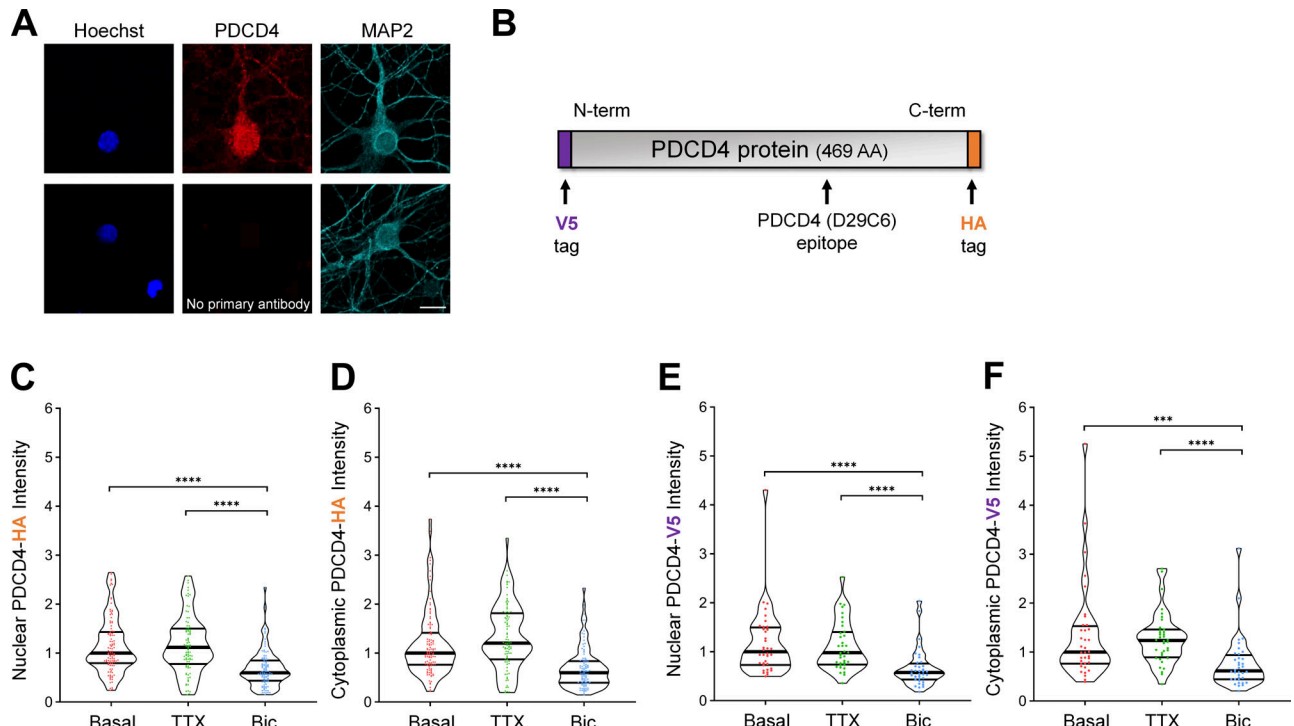

Figure S2. **Neuronal stimulation decreases PDCD4 protein concentration in the nucleus and cytoplasm of neurons.** Related to Fig. 2. **(A)** Top: ICC of endogenous PDCD4 protein. Bottom: Negative control (no primary PDCD4 antibody). Scale bar, 10 µm. **(B)** Schematic of PDCD4 protein, with locations of V5 tag, HA tag, and PDCD4 (D29C6) epitope (recognized by the PDCD4 Cell Signaling Technology antibody used in this study). **(C)** Violin plots of normalized nuclear HA ICC intensity in neurons transduced with PDCD4-HA AAV. Basal, $n$ = 107 cells; TTX, $n$ = 88 cells; Bic, $n$ = 102 cells from three sets of cultures. Basal median = 1.00; TTX median = 1.116; Bic median = 0.5972. Basal versus Bic, $P < 0.0001$; TTX versus Bic, $P < 0.0001$. **(D)** Violin plots of normalized cytoplasmic HA ICC intensity in the same cells as in C. Basal median = 1.00; TTX median = 1.203; Bic median = 0.5983. Basal versus Bic, $P < 0.0001$; TTX versus Bic, $P < 0.0001$. **(E)** Violin plots of normalized nuclear V5 ICC intensity in neurons transfected with V5-PDCD4 plasmid and cotransfected with GFP plasmid as a transfection marker. Basal, $n$ = 36 cells; TTX, $n$ = 36 cells; and Bic, $n$ = 36 cells from two sets of cultures. For each cell, the nuclear V5 intensity was normalized to the nuclear GFP intensity in order to normalize for differences in transfection efficiency between cells. Basal median = 1.00; TTX median = 0.9810; Bic median = 0.5760. Basal versus Bic, $P < 0.0001$; TTX versus Bic, $P < 0.0001$. **(F)** Violin plots of normalized cytoplasmic V5 ICC intensity in the same cells as in E. For each cell, the cytoplasmic V5 intensity was normalized to the nuclear GFP intensity in order to normalize for differences in transfection efficiency between cells. Basal median = 1.00; TTX median = 1.237; Bic median = 0.6167. Basal versus Bic, $P = 0.0002$; TTX versus Bic, $P < 0.0001$. ***, $P < 0.001$; ****, $P < 0.0001$; Mann-Whitney $U$ test with Bonferroni correction.

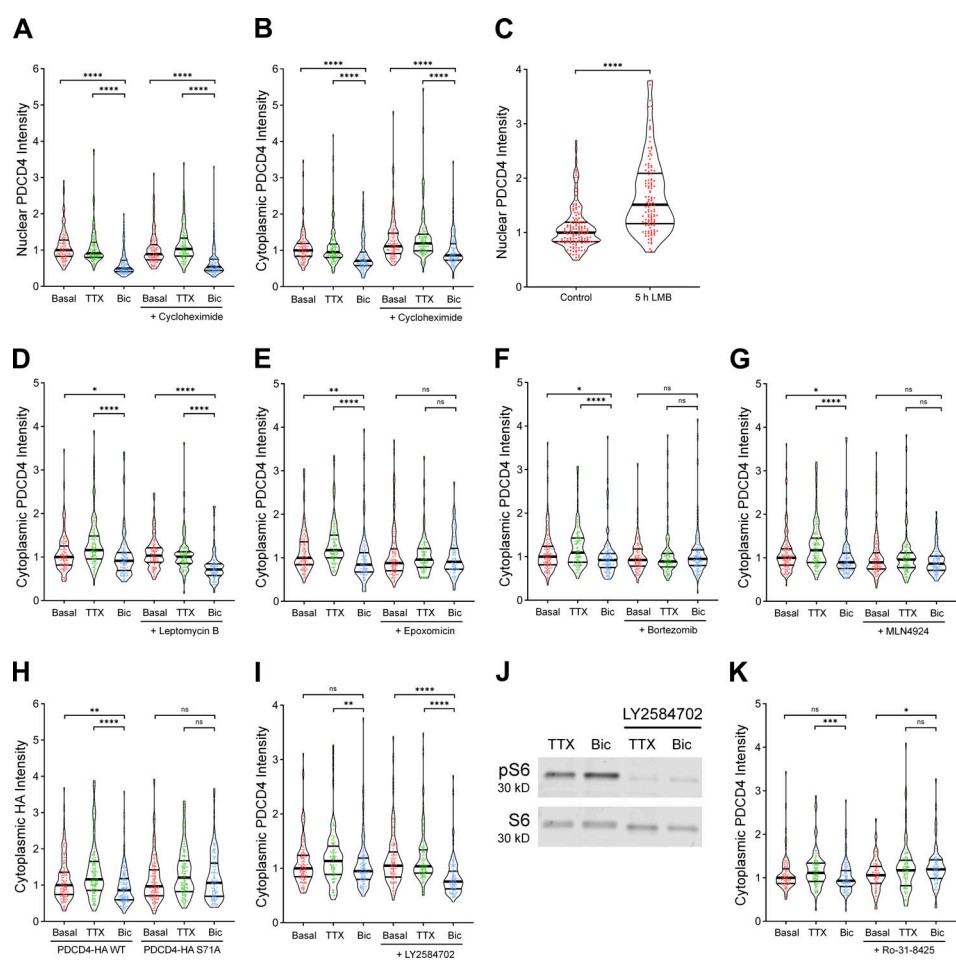

Figure S3. **Verification of nuclear export inhibitor LMB and S6K inhibitor Ly2584702 and quantification of cytoplasmic PDCD4.** Related to Figs. 3 and 4. **(A)** Violin plots of normalized nuclear PDCD4 ICC intensity. Basal, *n* = 117 cells; TTX, *n* = 118 cells; Bic, *n* = 109 cells; CHX-basal, *n* = 123 cells; CHX-TTX, *n* = 120 cells; and CHX-Bic, *n* = 104 cells from 3 sets of cultures. Basal median = 1.00; TTX median = 0.9119; Bic median = 0.4924; CHX-basal median = 0.8890; CHX-TTX median = 1.033; CHX-Bic median = 0.5375. Basal versus Bic, P < 0.0001; TTX versus Bic, P < 0.0001; CHX-basal versus CHX-Bic, P < 0.0001; CHX-TTX versus CHX-Bic, P < 0.0001. **(B)** Violin plots of normalized cytoplasmic PDCD4 ICC intensity in the same cells as in A. Basal median = 1.00; TTX median = 0.9501; Bic median = 0.7138; CHX-basal median = 1.112; CHX-TTX median = 1.191; CHX-Bic median = 0.8626. Basal versus Bic, P < 0.0001; TTX versus Bic, P < 0.0001; CHX-basal versus CHX-Bic, P < 0.0001; CHX-TTX versus CHX-Bic, P < 0.0001. **(C)** Violin plots of normalized nuclear PDCD4 ICC intensity. Control, *n* = 137 cells; and LMB, *n* = 122 cells from three sets of cultures. Control median = 1.00; LMB median = 1.513. Control versus LMB, P < 0.0001. **(D)** Violin plots of normalized cytoplasmic PDCD4 ICC intensity in the same cells as in Fig. 3 A. Basal median = 1.00; TTX median = 1.158; Bic median = 0.9127; LMB-basal median = 1.031; LMB-TTX median = 1.005; LMB-Bic median = 0.7122. Basal versus Bic, P = 0.034; TTX versus Bic, P < 0.0001; LMB-basal versus LMB-Bic, P < 0.0001; LMB-TTX versus LMB-Bic, P < 0.0001. **(E)** Violin plots of normalized cytoplasmic PDCD4 ICC intensity in the same cells as in Fig. 3 B. Basal median = 1.00; TTX median = 1.174; Bic median = 0.8439; Epox-basal median = 0.8789; Epox-TTX median = 0.9596; Epox-Bic median = 0.9077. Basal versus Bic, P = 0.001; TTX versus Bic, P < 0.0001; Epox-basal versus Epox-Bic, P = 1; Epox-TTX versus Epox-Bic, P = 0.8258. **(F)** Violin plots of normalized cytoplasmic PDCD4 ICC intensity in the same cells as in Fig. 3 C. Basal median = 1.00; TTX median = 1.093; Bic median = 0.9226; Bort-basal median = 0.9229; Bort-TTX median = 0.8904; Bort-Bic median = 0.9472. Basal versus Bic, P = 0.0156; TTX versus Bic, P < 0.0001; Bort-basal versus Bort-Bic, P = 1; Bort-TTX versus Bort-Bic, P = 0.3544. **(G)** Violin plots of normalized cytoplasmic PDCD4 ICC intensity in the same cells as in Fig. 3 D. Basal median = 1.00; TTX median = 1.173; Bic median = 0.8955; MLN-basal median = 0.8940; MLN-TTX median = 0.9603; MLN-Bic median = 0.8633. Basal versus Bic, P = 0.0324; TTX versus Bic, P < 0.0001; MLN-basal versus MLN-Bic, P = 0.6294; MLN-TTX versus MLN-Bic, P = 0.11. **(H)** Violin plots of normalized cytoplasmic HA ICC intensity in the same cells as in Fig. 4 C. WT-basal median = 1.00; WT-TTX median = 1.161; WT-Bic median = 0.8621; S71A-basal median = 0.9676; S71A-TTX median = 1.209; S71A-Bic median = 1.063. WT-basal versus WT-Bic, P = 0.0012; WT-TTX versus WT-Bic, P < 0.0001; S71A-basal versus S71A-Bic, P = 0.6704; S71A-TTX versus S71A-Bic, P = 0.2012. **(I)** Violin plots of normalized cytoplasmic PDCD4 ICC intensity in the same cells as in Fig. 4 E. Basal median = 1.00; TTX median = 1.14; Bic median = 0.95; LY-basal median = 1.05; LY-TTX median = 1.04; LY-Bic median = 0.76. Basal versus Bic, P = 0.4742; TTX versus Bic, P = 0.0088; LY-basal versus LY-Bic, P < 0.0001; LY-TTX versus LY-Bic, P < 0.0001. Note that, in these experiments, the basal versus Bic-induced decrease in cytoplasmic PDCD4 is not statistically significant for the untreated (no inhibitor) cells. The Bic-induced cytoplasmic decrease is smaller and less consistent than the nuclear decrease, with statistical significance in the control cells of Fig. S3, E–G, but not Fig. S3, I and K. Nonetheless, there is a significant Bic-induced decrease of cytoplasmic PDCD4 in the treated (LY2584702) cells, demonstrating that S6K is not required for the Bic-induced decrease of PDCD4. **(J)** Western blot of protein lysates from neurons treated with or without LY2584702. Western blot was stained with antibodies for phospho-S6 (Ser235/236) and total S6 to confirm that LY2584702 inhibits S6K activity. **(K)** Violin plots of normalized cytoplasmic PDCD4 ICC intensity in the same cells as in Fig. 4 F. Basal median = 1.00; TTX median = 1.114; Bic median = 0.9326; Ro-31-8425 (Ro)-basal median = 1.059; Ro-TTX median = 1.172; Ro-Bic median = 1.195. Basal versus Bic, P = 0.2592; TTX versus Bic, P = 0.0008; Ro-basal versus Ro-Bic, P = 0.0118; Ro-TTX versus Ro-Bic, P = 0.9892. *, P < 0.05; **, P < 0.01; ***, P < 0.001; and ****, P < 0.0001; Mann-Whitney *U* test with Bonferroni correction.

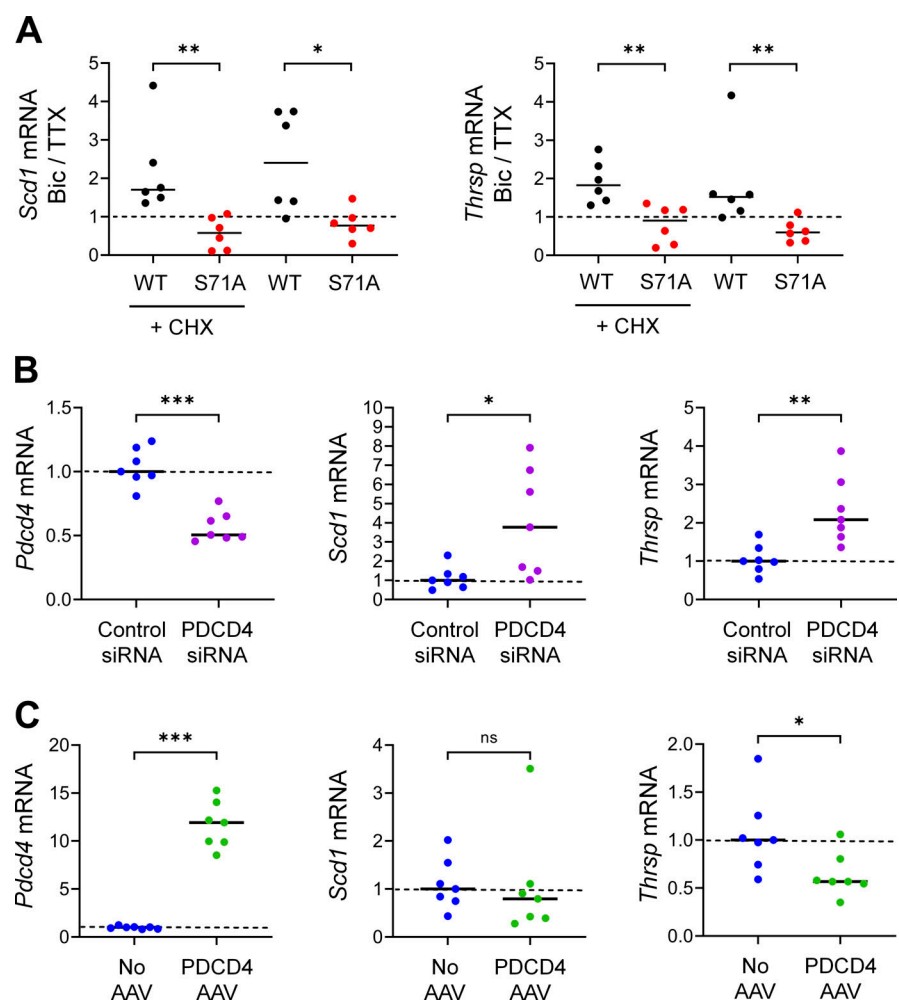

Figure S4.  **RT-qPCR of putative PDCD4 target genes.** Relating to Fig. 6. **(A)** RT-qPCR of putative PDCD4 target genes, *Scd1* and *Thrsp*, from TTX-silenced and Bic-stimulated neurons that were transduced with PDCD4 WT or S71A from six sets of cultures. Samples were normalized using two housekeeping genes, *Hprt* and *Gapdh*. The abundance of the target gene in each Bic sample was normalized to its respective TTX sample. *Scd1* WT CHX median = 1.702; *Scd1* S71A CHX median = 0.5776; *Scd1* WT median = 2.401; *Scd1* S71A median = 0.7672; *Thrsp* WT CHX median = 1.826; *Thrsp* S71A CHX median = 0.9080; *Thrsp* WT median = 1.522; *Thrsp* S71A median = 0.5995. *Scd1* CHX WT versus CHX S71A, P = 0.0022; *Scd1* WT versus S71A, P = 0.0260; *Thrsp* CHX WT versus CHX S71A, P = 0.0043; *Thrsp* WT versus S71A, P = 0.0043. **(B)** RT-qPCR of *Pdcd4*, *Scd1*, and *Thrsp* from PDCD4 knockdown experiments in TTX-silenced neurons from seven sets of cultures. Samples were normalized to the housekeeping gene *Gapdh*. All values were normalized to the median value of the control condition (control siRNA). *Pdcd4* control siRNA median = 1.00; *Pdcd4* PDCD4 siRNA median = 0.51. *Pdcd4* control versus PDCD4 siRNA, P = 0.0006. *Scd1* control siRNA median = 1.00; *Scd1* PDCD4 siRNA median = 3.77. *Scd1* control versus PDCD4 siRNA, P = 0.0111. *Thrsp* control siRNA median = 1.00; *Thrsp* PDCD4 siRNA median = 2.08. *Thrsp* control versus PDCD4 siRNA, P = 0.0023. **(C)** RT-qPCR of *Pdcd4*, *Scd1*, and *Thrsp* from PDCD4 overexpression experiments in TTX-silenced neurons from seven sets of cultures. Samples were normalized to the housekeeping gene *Gapdh*. All values were normalized to the median value of the control condition (no AAV). *Pdcd4* no AAV median = 1.00; *Pdcd4* PDCD4 AAV median = 11.93. *Pdcd4* no AAV versus PDCD4 AAV, P = 0.0006. *Scd1* no AAV median = 1.00; *Scd1* PDCD4 AAV median = 0.80. *Scd1* No AAV versus PDCD4 AAV, P = 0.3176. *Thrsp* no AAV median = 1.00; *Thrsp* PDCD4 AAV median = 0.57. *Thrsp* no AAV versus PDCD4 AAV, P = 0.0262. *, P < 0.05; **, P < 0.01; ***, P < 0.001; Mann-Whitney *U* test.

## Activity-dependent upregulated genes

### Promoters of putative PDCD4 target genes

| | Motif | % in targets: background | P-value |
|---|---|---|---|
| | JunD | 12:2 | 1E-4 |
| | TATA-Box | 42:22 | 1E-3 |
| | Klf4 | 30:13 | 1E-3 |
| | Klf5 | 66:46 | 1E-3 |
| | Sp1 | 36:20 | 1E-2 |
| | cJun | 15:5 | 1E-2 |
| | CRE | 15:5 | 1E-2 |
| | NFY | 31:18 | 1E-2 |
| | Isl1 | 45:29 | 1E-2 |
| | Atf2 | 15:6 | 1E-2 |

### Promoters of other CHX-insensitive genes

| | Motif | % in targets: background | P-value |
|---|---|---|---|
| | CRE | 23:6 | 1E-15 |
| | JunD | 13:2 | 1E-15 |
| | Atf1 | 30:12 | 1E-13 |
| | cJun | 16:5 | 1E-9 |
| | Atf7 | 21:8 | 1E-8 |
| | Atf2 | 17:6 | 1E-8 |
| | NFY | 33:17 | 1E-8 |
| | Sp1 | 36:22 | 1E-6 |
| | Klf5 | 61:46 | 1E-5 |
| | TATA-Box | 36:22 | 1E-5 |

## Activity-dependent downregulated genes

### Promoters of putative PDCD4 target genes

no significant motifs
(too few genes)

### Promoters of other CHX-insensitive genes

| | Motif | % in targets: background | P-value |
|---|---|---|---|
| | Nrf | 23:8 | 1E-6 |
| | Nrf1 | 20:8 | 1E-5 |
| | Sp1 | 34:18 | 1E-4 |
| | Gfy-Staf | 8:2 | 1E-3 |
| | Klf4 | 24:13 | 1E-3 |
| | Klf5 | 55:42 | 1E-2 |
| | Gfy | 6:2 | 1E-2 |
| | Elk4 | 30:20 | 1E-2 |

Figure S5. **Motif analysis of promoters of putative PDCD4 target genes.** Related to Fig. 6. Motif analyses of promoters of putative PDCD4 target genes (left column) and, for comparison, other CHX-insensitive activity-dependent genes (right column) using HOMER software. The top panel shows the motif image logos, enrichment, and P values for the top 10 motifs by P value for activity-dependent up-regulated genes, and the bottom panel shows the same but for activity-dependent down-regulated genes (only 8 motifs were significant for down-regulated genes).

**Provided online are five tables in Excel files. Table S1 contains the MS data from the nuclear proteomes of silenced and stimulated neurons. Table S2 lists the GO analysis of the APEX2-NLS MS. Table S3 contains the RNA-seq data from silenced and stimulated neurons, with PDCD4 WT or S71A. Table S4 lists the GO analysis for putative PDCD4 target genes. Table S5 lists the primer sequences used for cloning and RT-qPCR.**

