## [Peer Review File · The Journal of Cell Biology]

Neuronal activity regulates the nuclear proteome to promote activity-dependent transcription

Wendy Herbst, Weixian Deng, James Wohlschlegel, Jennifer Achiro, and Kelsey Martin

Corresponding Author(s): Kelsey Martin, David Geffen School of Medicine at UCLA and Jennifer Achiro, David Geffen School of Medicine at UCLA

Review Timeline:

Submission Date:	2021-03-24
Editorial Decision:	2021-04-21
Revision Received:	2021-08-23
Editorial Decision:	2021-09-08
Revision Received:	2021-09-12

Monitoring Editor: Louis Reichardt

Scientific Editor: Tim Spencer

Transaction Report:

DOI: <https://doi.org/10.1083/jcb.202103087>

April 21, 2021

Re: JCB manuscript #202103087

Dr. Kelsey C Martin
David Geffen School of Medicine at UCLA
BSRB 390B 615 Charles E Young Dr. S.
Los Angeles, California 90095

Dear Kelsey,

Thank you for submitting your manuscript entitled "Neuronal activity regulates the nuclear proteome to promote activity-dependent transcription". I am attaching to this decision letter the evaluations by two reviewers. Happily, both reviewers have overall positive assessments of the data and conclusions in this manuscript, so it appears to be one that will be acceptable after completion of recommended revisions.

The most important recommendation of the first reviewer is his/her point #1, a control experiment using puromycin or metabolic labelling to show that translation is blocked. The other items of concern identified by this reviewer appear to me to be quite easily addressable. The second reviewer has a somewhat more substantial list of concerns. I do agree with this referee's point #1 that demonstration of nuclear PDCD4's role vs. total PDCD4's role is important to establish. Similarly, I think that the experiment recommended in this reviewer's point #3 is important and straightforward. Point #4 is a recommendation for your consideration. Concerning this reviewer's point #2, I do not consider it essential to delve into PDCD4's mechanism beyond demonstrating a nuclear role. Concerning minor point #5, I do agree about the importance of demonstrating the nuclear location of APEX2-NLS in presence of activity and CHX. Minor point #6 seems to me to represent an extension beyond what is essential for this manuscript. I appreciate that it must be part of your future goals.

It seems to me to be quite likely that your lab members and you will be able to complete the few studies needed for a revision and return this revision for final consideration by the journal. I would recommend that the revision be accompanied by a copy in which changes are highlighted as well as a letter describing your changes and responses to reviewers. I would prefer to make a final decision without return of the revision to the reviewers, but may find it advisable to do this, depending upon my assessment of the manuscript's changes and the clarity and completeness of your responses to the reviewers' comments.

GENERAL GUIDELINES:

Text limits: Character count for an Article is < 40,000, not including spaces. Count includes title page, abstract, introduction, results, discussion, acknowledgments, and figure legends. Count does not include materials and methods, references, tables, or supplemental legends.

Figures: Articles may have up to 10 main text figures. Figures must be prepared according to the policies outlined in our Instructions to Authors, under Data Presentation, <https://jcb.rupress.org/site/misc/ifora.xhtml>. All figures in accepted manuscripts will be screened prior to publication.

IMPORTANT: It is JCB policy that if requested, original data images must be made available. Failure to provide original images upon request will result in unavoidable delays in publication. Please ensure that you have access to all original microscopy and blot data images before submitting your revision.

Supplemental information: There are strict limits on the allowable amount of supplemental data. Articles may have up to 5 supplemental figures. Up to 10 supplemental videos or flash animations are allowed. A summary of all supplemental material should appear at the end of the Materials and methods section.

As you may know, the typical timeframe for revisions is three to four months. However, we at JCB realize that the implementation of social distancing and shelter in place measures that limit spread of COVID-19 also pose challenges to scientific researchers. Lab closures especially are preventing scientists from conducting experiments to further their research. Therefore, JCB has waived the revision time limit. We recommend that you reach out to the editors once your lab has reopened to decide on an appropriate time frame for resubmission. Please note that papers are generally considered through only one revision cycle, so any revised manuscript will likely be either accepted or rejected.

Thank you for this interesting contribution to the Journal of Cell Biology. You can contact us at the journal office with any questions, cellbio@rockefeller.edu or call (212) 327-8588.

With best personal wishes,

Louis Reichardt, PhD
Editor, Journal of Cell Biology

Melina Casadio, PhD
Senior Scientific Editor, Journal of Cell Biology

Reviewer #1 (Comments to the Authors (Required)):

Herbst et al. present a modification of APEX2-proximity labeling that enables them to identify changes in the nuclear proteome of neurons following plasticity. The methodology could be expanded to other experimental paradigms and thus represents a useful resource for the wider community of biologists. Although, the authors provide immunofluorescence images showing

specific nuclear localization of the APEX2 construct, in order to better assess the specificity of their method they should perform GO-term analysis of the MS data collected. The authors identify the gene PDCD4 as a potential regulator of activity-mediated transcriptional changes. They then go on to show that PDCD4 undergoes proteasome-mediated degradation following PKC phosphorylation on residue S71. Lastly, by RNA-seq analyses on CHX-treated neurons carrying either WT or S71A mutant PDCD4, they are able to demonstrate a role for this factor in the transcriptional regulation of a subset of activity-dependent genes. We feel that the findings and the quality of the study are suitable for publication in the Journal of Cell Biology. However, in addition to GO term analysis of the MS data, we would ask the authors to address the following points:

- 1) The authors should perform puromycin labelling or another metabolic labeling method following 15 min of 60 μ M CHX treatment to show that translation is indeed blocked.
- 2) When performing differential expression analysis on Bic and TTX conditions they impose a positive $\log_2FC > 0.5$ for Bic and $\log_2FC < -0.5$ for TTX, what is the reason for this? Proteins could increase/decrease in abundance independent of the type of treatment.
- 3) In Figure S1A there is a 'No1 antibody' label, what does this refer to?
- 4) The authors stress that increases in glutamatergic transmission leads to a rapid and long-lasting reduction in PDCD4 abundance in the nucleus. However, this appears to be a global change and not nucleus specific. This should be clarified or the text should be modified.
- 5) The western blot presented in figure 4D does not reflect well the quantification shown, and a slight decrease in PDCD4 seems to take place despite S71A mutation. Have the authors tested other potential phosphorylation sites on PDCD4 that might be implicated in its degradation?
- 6) Albeit not the focus of the paper, the authors perform a number of experiments where they quantify changes in cytoplasmic PDCD4 levels and they observe a milder, but significant decrease following Bic treatment. However, in figure S3B this effect seems to be lost under control conditions and regained upon treatment with the S6K inhibitor LY2584702, which would argue that S6K is involved in cytoplasmic PDCD4 degradation. Could the authors discuss this?
- 7) When describing Figure 5A the authors should provide the fold change for the immediate early genes they describe (i.e. Npas4, Rgs2, Egr4), as the changes between the WT and S71A version of PDCD4 are hard to estimate visually.

Reviewer #2 (Comments to the Authors (Required)):

In this study, the authors investigated activity-dependent and translation-independent changes in the nuclear proteome of neurons. They performed a proximity ligation assay with the use of a nuclear form of APEX2 expressed in neuronal cultures isolated from rat forebrain and obtained 23 differentially labeled proteins in the presence (Bic treatment) or absence (TTX treatment) of neuronal activity. These proteins included PDCD4 in addition to positive controls such as CTRC1, HDAC4, and HDAC5. PDCD4 has already been shown to be degraded in response to mitogens through phosphorylation at Ser67/Ser71/Ser76 and in a manner dependent on beta-TRCP and the proteasome (Dorrello et al. 2006). The authors present results suggesting that phosphorylation of PDCD4 at Ser71 by PKC leads to its proteasome-dependent degradation within the nucleus also in

response to neuronal activity. Furthermore, they identified 91 neuronal activity-induced immediate-early genes (insensitive to CHX treatment) that were differentially expressed between cultures overexpressing either wild-type or S71A mutant forms of PDCD4.

Nuclear proteomics data for forebrain neurons with or without stimulation would be a valuable resource for readers in the fields of neuroscience and cell biology. The proposed role of PDCD4 in activity-induced gene regulation in neurons is unexpected and also would be of great interest. However, the authors should address the following points before publication of this paper.

Major points

1. The authors showed nuclear degradation of PDCD4 in response to neuronal activity but did not address the role of nuclear PDCD4 in activity-dependent transcription. "These findings suggest that the proteasome-mediated degradation of PDCD4 in the nucleus is important for regulating activity-dependent transcription following neuronal stimulation" (page 9) is thus an overstatement. Given that the role of nuclear proteins is a major focus of this paper (see the title), they should demonstrate that the loss of nuclear PDCD4 is indeed responsible for the activity-dependent transcription of its putative target genes.
2. Related to point 1, they should reveal the mechanism by which PDCD4 regulates transcription. This is especially important because of the known functions of PDCD4 in the cytoplasm.
3. The authors determined putative PDCD4 target genes by comparing the effects of overexpression of wild-type or S71A mutant forms of PDCD4. However, these genes may not be regulated by endogenous PDCD4. Does PDCD4 depletion affect transcription of these genes in resting neurons? Also, does overexpression of wild-type PDCD4 per se affect activity-induced transcription of these genes?
4. Characterization of the neuronal activity-dependent changes in the nuclear proteome is a highlight of this study. I therefore recommend that the authors show the obtained list (top 23 differentially labeled proteins) in Figure 1 to emphasize this point.

Minor points

5. It is necessary to confirm that the localization of APEX2-NLS is indeed confined to the nucleus in the presence of neuronal activity and CHX.
6. The impact of PDCD4-dependent transcription (in the presence of CHX) is limited to a small subset of activity-dependent genes. It would thus be desirable to show the functions of PDCD4-dependent transcription in neurons (such as in synapse formation, remodeling, and transmission) as proposed on page 9.

UNIVERSITY OF CALIFORNIA, LOS ANGELES

UCLA

BERKELEY • DAVIS • IRVINE • LOS ANGELES • MERCED • RIVERSIDE • SAN DIEGO • SAN FRANCISCO

SANTA BARBARA • SANTA CRUZ

OFFICE OF THE DEAN
DAVID GEFKEN SCHOOL OF MEDICINE AT UCLA
GEFFEN HALL
885 TIVERTON DRIVE, #400
LOS ANGELES, CALIFORNIA 90095-1722

Dr. Louis Reichardt
Editor, Journal of Cell Biology

August 22, 2021

Dear Dr. Reichardt,

We would like to thank you and the reviewers for their careful reading of our manuscript and for providing insightful feedback. In response, we conducted several new experiments and made a number of revisions to our manuscript. Responding to the reviews significantly strengthened our manuscript, which we hope is now suitable for publication in the *Journal of Cell Biology*.

In this letter, we address the reviewers' concerns and recommendations, beginning with the major points, followed by our responses to the minor points. We note that, as outlined in our responses below, we successfully conducted and/or addressed all of the experiments identified as most important by you as the editor.

Major Points

1. Reviewer #1 suggested that we perform metabolic labeling following cycloheximide (CHX) treatment to show that translation is effectively blocked in our experiments; this experiment was also identified by the editor as being critical. This is an important point because CHX is used in several experiments throughout the manuscript. To address this concern, we conducted a new experiment using AHA labeling to measure the amount of protein synthesis in the presence and absence of CHX (using the same 60 μ M concentration and 15 minute pre-incubation time used in our manuscript). We found that application of CHX strongly inhibited protein synthesis (**Fig S1A-B**). In support of this finding, we also demonstrate that CHX completely blocks the activity-dependent translation of FOS, a robust activity-dependent protein (**Fig S1C-D**, previously Fig S2C).
2. Reviewer #2 recommended that we demonstrate PDCD4's role in regulating transcription of its putative target genes through knockdown and overexpression in unstimulated neurons; the editor also underscored the importance of this concern. To address this recommendation, we conducted new experiments in which we compared expression of two putative PDCD4 transcriptional targets in neuronal cultures with siRNA-mediated knockdown of PDCD4, PDCD4 overexpression, or endogenous PDCD4 (**Fig S4B-C**). We found that indeed, compared to endogenous PDCD4, knockdown of PDCD4 resulted in increased expression of the target genes, demonstrating that endogenous PDCD4 does repress these target genes. PDCD4 overexpression had a smaller effect on the target genes, significantly decreasing one target gene and slightly (but non-significantly) decreasing the other target gene.
3. Reviewer #2 pointed out that the sentence, "These findings suggest that the proteasome-mediated degradation of PDCD4 in the nucleus is important for regulating activity-dependent transcription

following neuronal stimulation” (lines 393-4) is an overstatement because our experiment did not manipulate specifically nuclear PDCD4. The editor also noted the importance of this comment. We do not have a method to manipulate specifically nuclear PDCD4 (and not cytoplasmic PDCD4) because the activity-dependent decrease of PDCD4 is regulated by similar mechanisms in both the nucleus and the cytoplasm. We have thus removed the phrase “in the nucleus” from several sentences (lines 262-4 & lines 393-4) and added additional clarifying statements regarding cytoplasmic and nuclear PDCD4 (lines 399-405, line 194, line 350).

Minor Points

1. Reviewer #2 noted that a list of the top differentially labeled proteins would help to emphasize the finding of neuronal activity-dependent changes in the nuclear proteome. We agree with this point, and have now included new tables to Figure 1, displaying the top Bic-enriched and TTX-enriched labeled proteins (**Fig 1 E-F**).
2. Reviewer #1 recommended we perform GO-term analysis of the MS data we collected, and we have now included these analyses in **Fig S1G-H, Table S2**. These figures show that most of the top enriched GO terms are components of the nucleus, and that most of the top enriched proteins in our study are included in the GO term “nucleus”.
3. Reviewer #2 suggested that we confirm that APEX2-NLS remains localized to the nucleus even in the presence of neuronal activity and CHX. We now include an additional figure where we demonstrate that APEX2-NLS is localized to the nucleus in the presence of BIC/TTX and CHX (**Fig S1E-F**).
4. Reviewer #1 commented that the western blot in **Fig 4D** did not reflect well the quantification shown. We have updated the western blot image to show a different replicate, with values closer to the median value in the quantification.
5. Reviewer #1 noted that we did not observe a Bic-induced decrease in cytoplasmic PDCD4 in the control cells in **Fig S3I** (previously Fig S3B) and posited that this may implicate S6K in cytoplasmic PDCD4 degradation. As the reviewer mentioned, we often observe milder and less consistent activity-dependent decreases of cytoplasmic compared to nuclear PDCD4, and we have now revised several sentences in the text to point out the findings in the cytoplasm (lines 1065-1071, line 194, line 350). Nonetheless, there is a Bic-induced decrease of PDCD4 in cells treated with a S6K inhibitor, demonstrating that S6K is not required for the Bic-induced decrease of cytoplasmic PDCD4.
6. Reviewer #2 suggested that we perform experiments to reveal the mechanisms by which nuclear PDCD4 regulates transcription and its effect on synaptic function. The editor agrees these experiments are beyond the scope of this manuscript; however we do look forward to examining this in the future.
7. Reviewer #1 recommended that we provide the fold change values for the immediate early genes we mention when describing **Fig 5A**. We have now included these values in the text of the results (line 279) and note that the log₂FC values for all genes are listed in **Table S3** (previously Table S2).
8. Reviewer #1 found the descriptions of the log₂FC cutoffs for the MS experiment unclear, leading the reviewer mistakenly to conclude that we included changes in protein abundance in only certain directions depending on treatment. We have now simplified the text to clarify that the cutoff was log₂FC > 0.5 or < -0.5 (line 155).

9. Reviewer #1 found the label of “No 1° antibody” in **Fig S2A** (previously Fig S1A) unclear, so we have now updated the figure to include the word “primary” in place of “1°”.

Again, we thank the reviewers and editor for their critiques and suggestions. We are particularly pleased by Reviewer #2’s assessment that “Nuclear proteomics data for forebrain neurons with or without stimulation would be a valuable resource for readers in the fields of neuroscience and cell biology. The proposed role of PDCD4 in activity-induced gene regulation in neurons is unexpected and also would be of great interest.” The relevance of our study and experimental results to both the fields of neuroscience and cell biology is precisely why we chose to submit our manuscript to the *Journal of Cell Biology*.

Sincerely,

Kelsey C. Martin

Jennifer M. Achiro

September 8, 2021

RE: JCB Manuscript #202103087R

Dr. Kelsey C Martin
David Geffen School of Medicine at UCLA
BSRB 390B 615 Charles E Young Dr. S.
Los Angeles, California 90095

Dear Dr. Martin:

Thank you for submitting your revised manuscript entitled "Neuronal activity regulates the nuclear proteome to promote activity-dependent transcription". The manuscript has been assessed again by the original reviewers who both now recommend acceptance. Therefore, we would be happy to publish your paper in JCB pending final revisions necessary to meet our formatting guidelines (see details below).

A. MANUSCRIPT ORGANIZATION AND FORMATTING:

Full guidelines are available on our Instructions for Authors page, <https://jcb.rupress.org/submission-guidelines#revised>. **Submission of a paper that does not conform to JCB guidelines will delay the acceptance of your manuscript.**

1) Text limits: Character count for Articles is < 40,000, not including spaces. Count includes title page, abstract, introduction, results, discussion, and acknowledgments. Count does not include materials and methods, figure legends, references, tables, or supplemental legends. You are below this limit at the moment but please bear it in mind when revising.

2) Figure formatting: Scale bars must be present on all microscopy images, including inset magnifications. Molecular weight or nucleic acid size markers must be included on all gel electrophoresis.

3) Statistical analysis: Error bars on graphic representations of numerical data must be clearly described in the figure legend. The number of independent data points (n) represented in a graph must be indicated in the legend. Statistical methods should be explained in full in the materials and methods. For figures presenting pooled data the statistical measure should be defined in the figure legends. Please also be sure to indicate the statistical tests used in each of your experiments (both in the figure legend itself and in a separate methods section) as well as the parameters of the test (for example, if you ran a t-test, please indicate if it was one- or two-sided, etc.). Also, if you used parametric tests, please indicate if the data distribution was tested for normality (and if so, how). If not, you must state something to the effect that "Data distribution was assumed to be normal but this was not formally tested."

4) Materials and methods: Should be comprehensive and not simply reference a previous

publication for details on how an experiment was performed. Please provide full descriptions (at least in brief) in the text for readers who may not have access to referenced manuscripts. The text should not refer to methods "...as previously described."

5) Please be sure to provide the sequences for all of your primers/oligos and RNAi constructs in the materials and methods. You must also indicate in the methods the source, species, and catalog numbers (where appropriate) for all of your antibodies.

6) Microscope image acquisition: The following information must be provided about the acquisition and processing of images:

a. Make and model of microscope

b. Type, magnification, and numerical aperture of the objective lenses

c. Temperature

d. imaging medium

e. Fluorochromes

f. Camera make and model

g. Acquisition software

h. Any software used for image processing subsequent to data acquisition. Please include details and types of operations involved (e.g., type of deconvolution, 3D reconstitutions, surface or volume rendering, gamma adjustments, etc.).

7) References: There is no limit to the number of references cited in a manuscript. References should be cited parenthetically in the text by author and year of publication. Abbreviate the names of journals according to PubMed.

8) Supplemental materials: There are strict limits on the allowable amount of supplemental data. Articles may have up to 5 supplemental figures. At the moment, you meet this limit but please bear it in mind when revising.

Please also note that tables, like figures, should be provided as individual, editable files. A summary of all supplemental material should appear at the end of the Materials and methods section.

9) Conflict of interest statement: JCB requires inclusion of a statement in the acknowledgements regarding competing financial interests. If no competing financial interests exist, please include the following statement: "The authors declare no competing financial interests." If competing interests are declared, please follow your statement of these competing interests with the following statement: "The authors declare no further competing financial interests."

10) A separate author contribution section is required following the Acknowledgments in all research manuscripts. All authors should be mentioned and designated by their first and middle initials and full surnames. We encourage use of the CRediT nomenclature (<https://casrai.org/credit/>).

11) ORCID IDs: ORCID IDs are unique identifiers allowing researchers to create a record of their various scholarly contributions in a single place. At resubmission of your final files, please consider providing an ORCID ID for as many contributing authors as possible.

B. FINAL FILES:

Please upload the following materials to our online submission system. These items are required prior to acceptance. If you have any questions, contact JCB's Managing Editor, Lindsey Hollander

(lhollander@rockefeller.edu).

Thank you for your attention to these final processing requirements. Please revise and format the manuscript and upload materials within 7-14 days. If complications arising from measures taken to prevent the spread of COVID-19 will prevent you from meeting this deadline (e.g. if you cannot retrieve necessary files from your laboratory, etc.), please let us know and we can work with you to determine a suitable revision period.

Please contact the journal office with any questions, cellbio@rockefeller.edu.

Thank you for this interesting contribution, we look forward to publishing your paper in Journal of Cell Biology.

Sincerely,

Louis Reichardt, PhD
Senior Editor
Journal of Cell Biology

Tim Spencer, PhD
Executive Editor
Journal of Cell Biology

Reviewer #1 (Comments to the Authors (Required)):

I have no remaining concerns and think the paper should now be accepted at JCB.

Reviewer #2 (Comments to the Authors (Required)):

The authors have appropriately addressed my criticisms.